# ACCELERATED PREDICTIVE CODING NETWORKS VIA DIRECT KOLEN–POLLACK FEEDBACK ALIGNMENT

## ABSTRACT

Backpropagation (BP) is the cornerstone algorithm for training artificial neural networks, yet its reliance on update-locked global error propagation limits biological plausibility and hardware efficiency. Predictive coding (PC), originally proposed as a model of the visual cortex, relies on local updates that allow parallel learning across layers. However, practical implementations face two key limitations: error signals must still propagate from the output to early layers through multiple inference-phase steps, and feedback decays exponentially during this process, leading to vanishing updates in early layers. These issues restrict the efficiency and scalability of PC, undermining its theoretical advantage in parallelization over BP. We propose direct Kolen–Pollack predictive coding (DKP-PC), which simultaneously addresses both feedback delay and exponential decay, yielding a more efficient and scalable variant of PC while preserving update locality. Leveraging the direct feedback alignment and direct Kolen–Pollack algorithms, DKP-PC introduces learnable feedback connections from the output layer to all hidden layers, establishing a direct pathway for error transmission. This yields an algorithm that reduces the theoretical error propagation time complexity from $\mathcal{O}(L)$, with $L$ being the network depth, to $\mathcal{O}(1)$, enabling parallel updates of the parameters. Moreover, empirical results demonstrate that DKP-PC achieves performance at least comparable to, and often exceeding, that of standard PC, while offering improved latency and computational performance. By enhancing both scalability and efficiency of PC, DKP-PC narrows the gap between biologically-plausible learning algorithms and BP, and unlocks the potential of local learning rules for hardware-efficient implementations.

## 1 INTRODUCTION

Major advances in artificial intelligence, from image recognition (LeCun et al., 2002; Krizhevsky et al., 2017; Alom et al., 2018) to image generation (Kingma & Welling, 2013; Parmar et al., 2018; Goodfellow et al., 2020) and natural language processing (Hochreiter & Schmidhuber, 1997; Vaswani et al., 2017; Beck et al., 2024), have all been enabled by backpropagation of error (BP), the fundamental algorithm underlying the training of artificial neural networks (ANNs) (Linnainmaa, 1970; Rumelhart et al., 1986; Werbos, 1988). However, several studies have put into question the plausibility of its direct implementation in biological neural systems (Grossberg, 1987; Lillicrap et al., 2016; Lillicrap & Santoro, 2019; Whittington & Bogacz, 2019; Ellenberger et al., 2024). Two primary concerns come from (i) the reliance on a global error signal that must be propagated backward and sequentially through the network hierarchy, thereby blocking parameter updates, and (ii) early layers depending directly on error signals generated by distant nodes. These biological plausibility issues of BP are commonly referred to as *update locking* and *non-locality* (Nøkland, 2016; Frenkel et al., 2021; Ororbia, 2023). Importantly, they lead to inefficiencies in hardware implementations, imposing memory and latency overheads (Mostafa et al., 2018; Frenkel et al., 2023).

Predictive coding (PC), originally introduced as a model of the visual cortex in the human brain (Rao & Ballard, 1999; Huang & Rao, 2011), is emerging as a promising alternative to BP, potentially alleviating its update-locking and non-locality limitations (Millidge et al., 2022a; Salvatori et al., 2023). Its framework is grounded in Bayesian inference under the Free Energy Principle (Friston, 2005; Friston et al., 2006; Friston & Kiebel, 2009), providing a rigorous mathematical foundation with connections to information theory (Elias, 1955; 2003) and energy-based models (Millidge et al.,

2022b;c). Rather than minimizing a global error signal, PC minimizes the network's *variational free energy* (FE), defined as the sum of layer-wise squared errors between each layer's activity and its incoming prediction. Unlike BP, where weights are directly updated, PC learning has two phases. In the *inference phase*, neural activity is updated to minimize the FE, and in the *learning phase*, weights are updated based on the optimized neural activity. However, while this framework yields local and layer-wise update rules, the error in PC is still generated at the output and must propagate backward during inference. This error-delay limitation causes PC to be significantly slower than BP, and limits its efficiency and suitability for custom hardware implementations (Zahid et al., 2023). Moreover, the delayed error decays exponentially with depth, yielding vanishing updates in early layers (Pinchetti et al., 2024; Goemaere et al., 2025).

To address these limitations, we propose to propagate error information from the output layer to all hidden layers, yielding an instantaneous error term across the hierarchy. We thus build on feedback alignment methods (Lillicrap et al., 2014). Direct feedback alignment (DFA) (Nøkland, 2016) uses random direct feedback connections to deliver error signals from the output to all hidden layers, avoiding both error delay and decay. However, DFA scales poorly, especially in deep convolutional networks. Direct Kolen-Pollack (DKP) improves DFA by learning the feedback matrices (Webster et al., 2020), incorporating learning rules inspired by the Kolen-Pollack (KP) algorithm (Kolen & Pollack, 1994; Akrout et al., 2019), thereby enhancing performance while preserving locality. Figure 1 illustrates these frameworks and shows how our proposed direct KP predictive coding (DKP-PC) integrates primitives of both PC and DKP.

Our contributions are summarized as follows:

1. We extend the empirical analysis of Webster et al. (2020) by providing a mathematical motivation for why DKP achieves closer alignment with BP than standard DFA. This novel view further supports the integration of DKP within the PC framework as an efficient preliminary weight update to generate an instantaneous error term at every layer.

2. We introduce the DKP-PC algorithm, which simultaneously mitigates the feedback error delay and exponential decay limitations of BP while preserving locality. This, for the first time, enables full parallelization in PC networks regardless of batch size. We further discuss how our proposed PC variant achieves a time complexity of $\mathcal{O}(1)$, compared to $\mathcal{O}(L)$ for BP, with $L$ being the network depth.

3. We provide a theoretical and empirical analysis of the synergy between DKP-PC components, demonstrating how the PC neural activity update, under the DKP regime, leads to improved feedback matrix update and, ultimately, to better and more stable gradient alignment with BP compared to standard DKP.

4. We empirically demonstrate that DKP-PC performs on par with, or outperforms, both DKP and PC, benchmarking them across fully-connected and convolutional networks up to VGG-9 on Tiny ImageNet. We further assess DKP-PC's computational efficiency, showing that it consistently achieves more than a $60\%$ reduction in training time for both VGG-7 and VGG-9.

## 2 BACKGROUND

In this section, we review from a mathematical perspective the core concepts of BP, DFA/DKP, and PC, which form the basis of our DKP-PC algorithm.

### 2.1 BACKPROPAGATION

BP enables recursive and efficient computation of parameter gradients by applying the chain rule of calculus to propagate error derivatives from the output layer back through the network (Linnainmaa, 1970; Rumelhart et al., 1986). Let us consider a neural network as shown in Figure 1(A), where each layer $\ell \in \{0, \ldots, L\}$ is associated with an activity vector $x_\ell \in \mathbb{R}^{d_\ell}$, where $x_0$ denotes the input and $x_L$ the output, and $d_\ell$ is the number of neurons in layer $\ell$. The forward pass is defined recursively as

$$z_\ell = \Theta_{\ell-1} x_{\ell-1}, \quad x_\ell = f(z_\ell), \quad 1 \le \ell \le L, \tag{1}$$

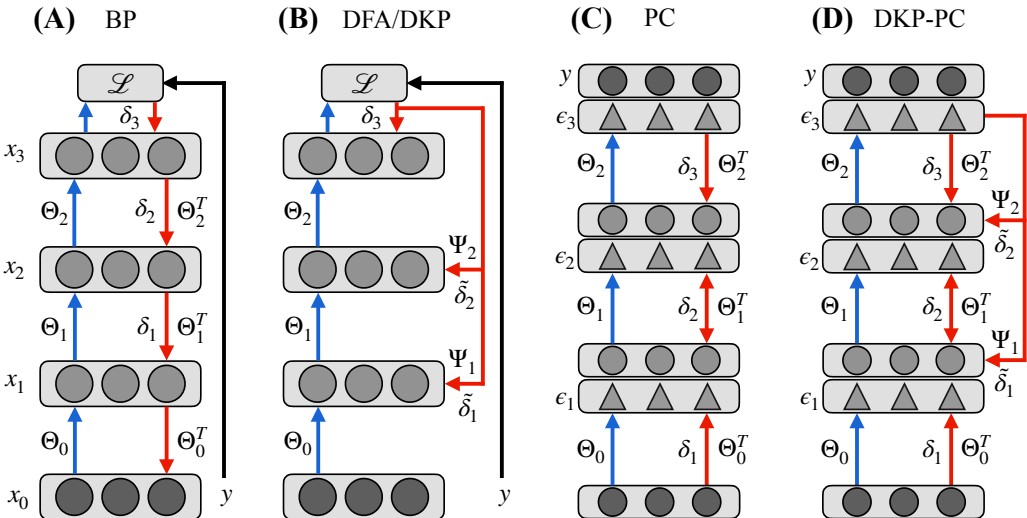

Figure 1: DKP-PC embeds DKP within the PC framework to address the error feedback delay and exponential decay issues of PC. Blue arrows represent forward connections, red arrows represent feedback connections. Neural activities are shown as gray circles, with clamped values in darker gray; $x_0$ denotes the input, $y$ the target. $\mathcal{L}$ is the loss function, with $\delta_\ell$ the BP error, $\tilde{\delta}_\ell$ its approximations, and $\epsilon_\ell$ the PC error neurons, represented as triangles. (A) BP propagates the global error sequentially. (B) DFA and DKP propagate the error directly from the output to each layer. (C) PC minimizes local errors through an inference phase, followed by a learning phase that updates weights. (D) DKP-PC uses DKP's direct feedback to provide instantaneous error signals at all layers, accelerating the PC inference phase while preserving local weight updates.

where $\Theta_{\ell-1} \in \mathbb{R}^{d_\ell \times d_{\ell-1}}$ is the synaptic weight matrix mapping activity at layer $\ell - 1$ to layer $\ell$, and $f : \mathbb{R}^{d_\ell} \to \mathbb{R}^{d_\ell}$ is typically an element-wise non-linear activation function. The output error is then expressed in terms of the least-squared error (LSE)

$$\mathcal{L} = \frac{1}{2}\|x_L - y\|_2^2, \tag{2}$$

where $x_L$ is the network's output and $y \in \mathbb{R}^{d_L}$ is the target vector. Applying the chain rule, the recursively backpropagated errors $\frac{\partial \mathcal{L}}{\partial z_\ell} = \delta_\ell \in \mathbb{R}^{d_\ell}$ are thus

$$\delta_\ell = \begin{cases} x_L - y & \text{if } \ell = L, \\ f'(z_\ell) \odot (\Theta_\ell^\top \delta_{\ell+1}) & \text{otherwise,} \end{cases} \tag{3}$$

where $\odot$ denotes the Hadamard product between the activation derivative $f'(z_\ell) \in \mathbb{R}^{d_\ell}$ and the error term[1]. The weights are then updated according to

$$\Delta\Theta_\ell = -\alpha\frac{\partial \mathcal{L}}{\partial \Theta_\ell} = -\alpha\big(\delta_{\ell+1}x_\ell^\top\big), \tag{4}$$

where $\alpha \in (0, 1)$ is the weight learning rate.

## 2.2 DIRECT KOLEN-POLLACK FEEDBACK ALIGNMENT

DFA explicitly addresses the challenge of iteratively backpropagating error information in BP, yielding a local and more biologically plausible algorithm (Nøkland, 2016). To achieve this, DFA introduces random matrices $\Psi_\ell \in \mathbb{R}^{d_\ell \times d_L}$ that connect the output layer directly to each hidden layer

---

[1]Formally, $f'(z_\ell) = \frac{\partial x_\ell}{\partial z_\ell} \in \mathbb{R}^{d_\ell \times d_\ell}$ is the Jacobian matrix of the activation function. However, for element-wise activations, this Jacobian is diagonal with entries $f'(z_\ell)$, so the matrix-vector multiplication simplifies to the Hadamard product.

in the network, as illustrated in Figure 1(B). These matrices enable the direct propagation of the error signal $\delta_L \in \mathbb{R}^{d_L}$ generated at the output layer, avoiding the iterative backward propagation in Eq. (3), by projecting it directly into each hidden layer as

$$\tilde{\delta}_\ell = f'(z_\ell) \odot (\Psi_\ell \delta_L), \tag{5}$$

thereby leading to the following local weight update rule:

$$\Delta\tilde{\Theta}_\ell = -\alpha\big(\tilde{\delta}_{\ell+1} x_\ell^\top\big). \tag{6}$$

DKP extends DFA by introducing a local learning rule for the feedback matrices $\Psi_\ell$ (Webster et al., 2020), inspired by the KP algorithm (Kolen & Pollack, 1994; Akrout et al., 2019). In contrast to DFA where the random matrices $\Psi_\ell$ are kept fixed, DKP updates them with

$$\Delta\Psi_\ell = -\alpha\big(x_\ell \delta_L^\top\big), \tag{7}$$

where the synaptic plasticity of $\Psi_\ell$ depends only on the connected hidden layer's activity and the output error signal. Note that this update depends only on local information on top of global error information that is shared across the network, and thus can be parallelized without update locking. In Appendix A.1, we extend the empirical analysis of Webster et al. (2020) by providing a mathematical demonstration that, under linear assumptions, the feedback matrices converge to values that incorporate the recursive chain in (4), despite the dimensionality mismatch between $\Psi_\ell$ and $\Theta_\ell$. This offers a new theoretical perspective explaining why DKP aligns more closely with BP compared to DFA, showing that it converges to a recursive Moore–Penrose pseudoinverse chain of the forward weights:

$$\Psi_{L-\ell} = \begin{cases} \Theta_{L-\ell}^\top & \text{if } \ell = 1, \\ \Theta_{L-\ell}^\top \big(\Psi_{L-\ell+1}^\top\big)^+ & \text{if } 1 < \ell < L. \end{cases} \tag{8}$$

## 2.3 PREDICTIVE CODING

PC models the brain as a Bayesian hierarchical generative model, in which latent variables represent the causes of sensory stimuli and are assumed to follow a Gaussian distribution (Rao & Ballard, 1999; Friston, 2005; Friston & Kiebel, 2009). Neural activities $x_\ell \in \mathbb{R}^{d_\ell} \sim \mathcal{N}(\mu_\ell, \Sigma_\ell)$ represent latent variables at layer $\ell$, with mean $\mu_\ell \in \mathbb{R}^{d_\ell}$ and covariance matrix $\Sigma_\ell \in \mathbb{R}^{d_\ell \times d_\ell}$. As done by several works in literature, we assume the generative model's covariance to be fixed to the identity matrix $\Sigma_\ell = I$ (Pinchetti et al., 2022; Millidge et al., 2022a; Salvatori et al., 2024). The distribution's mean $\mu_\ell$ is parametrized by the previous layer's state through the synaptic weights $\Theta_\ell \in \mathbb{R}^{d_\ell \times d_{\ell-1}}$ connecting them, according to the relation $\mu_\ell = f(\Theta_{\ell-1} x_{\ell-1})$, where $f : \mathbb{R}^{d_\ell} \to \mathbb{R}^{d_\ell}$ is a non-linear mapping. The joint generative model over all $L + 1$ latent-variables layers is

$$p(x_0, \ldots, x_L; \Theta_0, \ldots, \Theta_{L-1}) = \mathcal{N}\big(x_0; \mu_0, \Sigma_0\big) \prod_{\ell=1}^{L} \mathcal{N}\big(x_\ell; f(\Theta_{\ell-1} x_{\ell-1}), \Sigma_\ell\big), \tag{9}$$

where $x_0$ and $x_L$ are clamped respectively to the input and target vectors, in classification settings. The exact posterior distribution inference $p(x_0, \ldots, x_{L-1} \mid x_L)$ is generally intractable (Friston, 2005), so PC employs variational inference to approximate the latter with a tractable distribution $q(x_0, \ldots, x_L)$, defined as

$$q(x_0, \ldots, x_{L-1}) = \prod_{\ell=0}^{L-1} q(x_\ell), \tag{10}$$

where $q(x_\ell) \in \mathbb{R}^{d_\ell}$ is the variational distribution over the layer $\ell$. Following PC literature for classification tasks, we model the variational posterior as a Dirac delta centered on parameter $\phi_\ell$:

$$q(x_\ell; \phi_\ell) = \delta(x_\ell - \phi_\ell), \tag{11}$$

where $\phi_\ell \in \mathbb{R}^{d_\ell}$ approximates $x_\ell$, which corresponds to the mode of the true posterior (Bogacz, 2017; Millidge et al., 2021; Pinchetti et al., 2022; Salvatori et al., 2024). This formulation provides a deterministic approximation of the latent variables' mode. Variational inference reduces the problem of maximizing Eq. (9) to minimizing the Kullback–Leibler divergence between the variational

and true posterior distributions $D_{KL}(q \parallel p)$. Equivalently, this corresponds to minimizing the variational FE, which constitutes an upper bound on $D_{KL}(q \parallel p)$ (Friston & Kiebel, 2009). Under the assumptions of identity covariance matrices for the generative model and a Dirac delta variational posterior, the FE can be expressed as

$$F = \frac{1}{2} \sum_{\ell=1}^{L} \|\epsilon_\ell\|_2^2, \tag{12}$$

where the prediction errors $\epsilon_\ell \in \mathbb{R}^{d_\ell}$ at layer $\ell$ are defined as

$$\epsilon_\ell = \phi_\ell - f(\Theta_{\ell-1}\phi_{\ell-1}), \tag{13}$$

and are considered as dedicated units, represented by triangles in Figure 1(C). In prediction tasks, $\phi_0$ is not inferred, as the input layer is clamped to the input vector, and the network is initialized via a forward pass as in Eq. (1). After this initialization, the output layer $\phi_L$ is clamped to the target vector, so it is also not inferred. Minimizing Eq. (12) results in local updates for both neurons and weights, enabling layer-wise learning.

PC learning is divided into two phases: the inference and the learning phases. During the inference phase, Eq. (12) is optimized with respect to the variational parameters $\phi_\ell$, updating the neural activities iteratively, whereas during the learning phase, the synaptic parameters $\Theta_\ell$ are updated using the resulting neural configuration to minimize the same objective. This yields the neural activity update

$$\Delta\phi_\ell = -\gamma \frac{\partial F}{\partial \phi_\ell} = \gamma \Big( (f'(\Theta_\ell \phi_\ell) \odot \Theta_\ell)^\top \epsilon_{\ell+1} - \epsilon_\ell \Big), \tag{14}$$

where $\gamma \in (0,1)$ is the neural activity learning rate. The rule is local, as the activity of $\phi_\ell$ is influenced only by the adjacent error nodes $\epsilon_\ell$ and $\epsilon_{\ell+1}$. After the inference phase is completed, the synaptic connections are updated using the final neural activity configuration $\phi_\ell^*$, following

$$\Delta\Theta_\ell = -\alpha \frac{\partial F}{\partial \Theta_\ell} = \alpha \left( f'(\Theta_\ell \phi_\ell^*) \odot \epsilon_{\ell+1} \phi_\ell^{*\top} \right), \tag{15}$$

where $\alpha \in (0,1)$ is the weight learning rate. The weight update is local, as it depends only on the error neurons of the next layer $\epsilon_{\ell+1}$, and on the optimized neural activity of the current layer $\phi_\ell^*$.

## 3 METHODOLOGY

In this section, we first introduce the issues of error delay and error decay in PC, and subsequently demonstrate how the proposed DKP-PC algorithm provides a unified solution to both of them.

### 3.1 FEEDBACK ERROR DELAY AND DECAY

Let us consider a forward-initialized PC network with $L+1$ latent variables layers, whose neural dynamics evolve in discrete time steps $t \in \mathbb{N}_0$ according to Eq. (14), explicitly denoting the time dependence of neural activities $\phi_\ell(t)$ and prediction errors $\epsilon_\ell(t)$ during the inference phase.

*Error propagation delay* – At the initial time $t = 0$, the neural activity of each layer corresponds to the prediction from the previous one, resulting in null error values at every layer in the network:

$$\phi_\ell(0) = f(\Theta_{\ell-1}\phi_{\ell-1}(0)) \implies \epsilon_\ell(0) = \phi_\ell(0) - f(\Theta_{\ell-1}\phi_{\ell-1}(0)) = 0. \tag{16}$$

The network is thus at equilibrium, since the FE in Eq. (12) is minimized (Whittington & Bogacz, 2017). Assuming an incorrect prediction, clamping the target vector $y$ to the output layer at $t = 0$ induces a non-zero error at the final layer $\epsilon_L(0) \neq 0$, as shown in Figure 2(A). Since each layer updates its activity based on its own and the subsequent layer's error (see Eq. (14)), the error propagates backward at most one layer per time step. Thus, the error takes $\hat{t} = L - \ell$ inference-phase steps to reach layer $\ell$ (Zahid et al., 2023) (see theorem and proof in Appendix A.2).

*Error exponential decay* – The error at optimization time $\hat{t} = L - \ell$, denoted $\epsilon_\ell(\hat{t})$, decays exponentially as it propagates backwards through the network, as shown in Figure 2(A). The decay rate is determined by both the learning rate and the distance from the output layer (Goemaere et al., 2025),

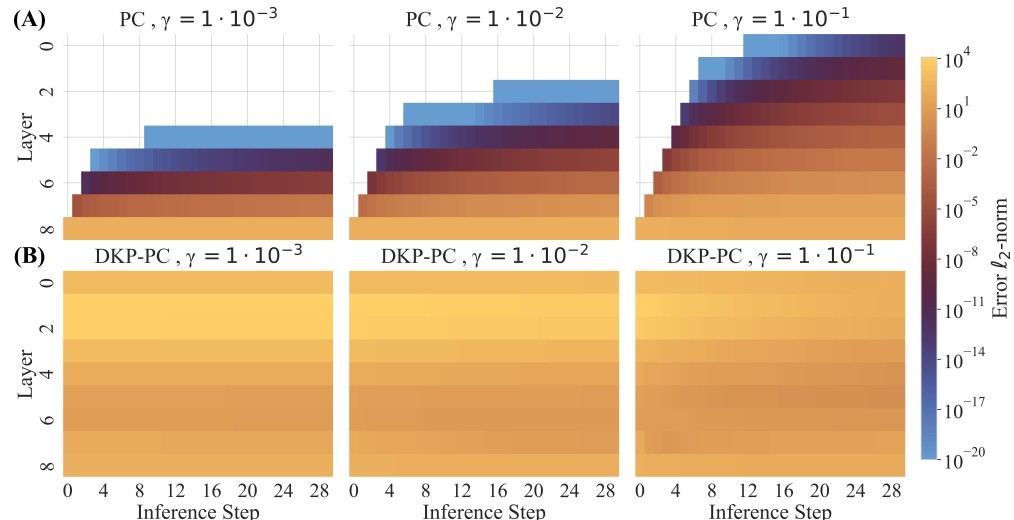

Figure 2: Error propagation in PC (A) and DKP-PC (B) during the inference phase of a VGG-9 network trained on a single CIFAR-10 batch, at different magnitudes of the neural activity learning rate $\gamma$. In (A), PC exhibits both an error decay problem, where the error magnitude decreases exponentially with network depth, and an error delay problem, as the error signal flows through the network sequentially, undermining the theoretical parallelism. White colour represents values equal to zero or below the numerical precision. In (B), DKP-PC mitigates both issues, generating a more uniform error signal across all layers at the start of neural activity optimization.

and the squared $\ell_2$-norm $\|\epsilon_\ell(\hat{t})\|_2^2$ is upper-bounded by a quantity $\propto \gamma^{2(L-\ell)}$ (see theorem and proof in Appendix A.3).

Both issues originate from the fact that the error is generated only at the output layer, with the delay arising from its layer-by-layer propagation through the network and the exponential decay resulting from its progressive reduction by the neural activity learning rate at each iteration.

### 3.2 Direct Kolen-Pollack Predictive Coding

We propose to introduce learnable feedback connections from the output layer to each hidden layer $\Psi_\ell \in \mathbb{R}^{d_\ell \times d_L}$, $\forall \ell \in \{1, \cdots, L-1\}$, to enable a preliminary update of the forward weights (Figure 1(D)). This approach, inspired by the DKP algorithm, not only induces approximate alignment with BP toward minimizing the output error (Appendix A.1), but also ensures that a non-zero error term is generated at every layer at the beginning of the inference phase, since the condition in Eq. (16) no longer holds (Figure 2(B)).

The resulting algorithm, denoted as DKP-PC, not only solves the error propagation delay and exponential decay issues of PC, but also allows speeding up the inference phase of PC. Indeed, after the error is directly propagated, we empirically show that a single inference-phase step is sufficient to match or even surpass the performance of standard PC, which typically requires a number of steps at least equal to, and often exceeding, the network depth (Pinchetti et al., 2024). We provide the pseudocode of DKP-PC in Algorithm 1, and formally outline the main steps below.

*Direct feedback alignment update* – After the forward initialization of the network, assuming a non-zero error at the output layer $\epsilon_L$, we perturb the equilibrium by taking a first weight update according to Eq. (6), using PC's last layer error neurons:

$$\tilde{\Theta}_\ell = \Theta_\ell + \Delta\tilde{\Theta}_\ell$$
$$= \Theta_\ell - \alpha\big(f'(\Theta_\ell\phi_\ell) \odot (\Psi_{\ell+1}\delta_L)\phi_\ell^\top\big) \tag{17}$$
$$= \Theta_\ell + \alpha\big(f'(\Theta_\ell\phi_\ell) \odot (\Psi_{\ell+1}\epsilon_L)\phi_\ell^\top\big),$$

which can be performed in parallel for each layer, as there is no recursive dependence.

---

**Algorithm 1** Direct Kolen-Pollack Predictive Coding (DKP-PC)

---

1: **for** each $(x, y) \in \mathcal{D}$ **do**
    *0) Forward initialization*
2:     $\phi_0 \leftarrow x$
3:     **for** $\ell = 1$ to $L - 1$ **do**                                      ▷ **Sequential**
4:         $\phi_\ell \leftarrow f(\Theta_{\ell-1}\phi_{\ell-1})$
5:     **end for**
6:     $\phi_L \leftarrow y$
7:     $\epsilon_L \leftarrow y - f(\Theta_{L-1}\phi_{L-1})$
    *1) Direct Feedback Alignment update*
8:     **for** $\ell = 0$ to $L - 1$ **do**                                          ▷ **Parallel**
9:         $\Theta_\ell \leftarrow \Theta_\ell + \alpha\big(f'(\Theta_\ell\phi_\ell) \odot (\Psi_{\ell+1}\epsilon_L)\phi_\ell^\top\big)$
10:    **end for**
    *2) Inference phase*
11:    **for** $t = 0$ to $T$ **do**                                           ▷ **T=1 for DKP-PC**
12:        **for** $\ell = 1$ to $L$ **do**                                  ▷ **Parallel**
13:           $\epsilon_\ell \leftarrow \phi_\ell - f(\Theta_{\ell-1}\phi_{\ell-1})$
14:           $\phi_\ell \leftarrow \phi_\ell - \gamma\frac{\partial F}{\partial\phi_\ell}$
15:        **end for**
16:    **end for**
    *3) Learning phase*
17:    **for** $\ell = 0$ to $L - 1$ **do**                                      ▷ **Parallel**
18:        $\Theta_\ell \leftarrow \Theta_\ell - \alpha\frac{\partial F}{\partial\Theta_\ell}$
19:    **end for**
    *4) Direct Kolen-Pollack update*
20:    **for** $\ell = 1$ to $L - 1$ **do**                                     ▷ **Parallel**
21:        $\Psi_\ell \leftarrow \Psi_\ell + \alpha\,\phi_\ell\epsilon_L^\top$
22:    **end for**
23: **end for**

---

*Inference phase* – After this update, an error term is generated at every layer since the first time step:

$$\|\epsilon_\ell(0)\|_2^2 = \|\phi_\ell(0) - f(\tilde{\Theta}_{\ell-1}\phi_{\ell-1}(0))\|_2^2 > 0, \tag{18}$$

where $\phi_\ell(0) = f(\Theta_{\ell-1}\phi_{\ell-1}(0))$ after forward initialization. This provides a non-zero error term instantaneously in every layer-wise component of the FE in Eq. (12). Consequently, every layer can independently update its neural activity according to Eq. (14), without performing null updates while waiting for the propagation of the error from the last layer. Thus, the single-step neural activity optimization performed takes the following form:

$$\Delta\phi_\ell = \gamma\Big((J_f(\tilde{\Theta}_\ell\phi_\ell)\,\tilde{\Theta}_\ell)^\top \epsilon_{\ell+1} - \epsilon_\ell\Big). \tag{19}$$

Furthermore, in contrast to standard PC, the neural activity now incorporates the information injected into the forward weights by the preliminary DKP update. In Appendix A.4.1, we show theoretically in Eq. (64) that, under linear assumptions, this corresponds to enforcing alignment and regularization through the single-step neural activity update, which in turn improves the alignment of the forward and feedback weights, as further supported by empirical evidence in Appendix A.4.2. Lastly, although the update in Eq. (19) is applied only once to maximize the acceleration of PC networks afforded by DKP-PC, our method is not limited to this setting. As further discussed in Appendix A.4.3, DKP-PC can leverage the same trade-off as standard PC networks, it can further minimize the network's FE to achieve higher classification accuracy by making use of multiple inference steps, at the cost of increased training time.

*Learning phase and DKP update* – After this single local update, both feedforward and feedback weight matrices are updated, which can be fully parallelized. Starting from the feedforward weight matrices $\Theta_\ell$, the update rule is unchanged from the standard PC update in Eq. (15), with the difference of using the neural activity resulting from Eq. (19). Feedback weight matrices $\Psi_\ell$ now also incorporate PC's optimized neural activity:

$$\Delta\Psi_\ell = -\alpha\big(\phi_\ell^*\delta_L^\top\big) = \alpha\big(\phi_\ell^*\epsilon_L^\top\big). \tag{20}$$

With DKP-PC, we introduce the first PC variant that is fully parallelizable. Although its sequential formulation between stages may appear to challenge parallelizability and strict biological plausibility, each stage relies exclusively on locally available variables. This enables full parallelization across layers and preserves the locality of computation throughout the entire learning process, which is a hallmark of biologically plausible learning. As a result, the backward time complexity of the network is reduced from $\mathcal{O}(L)$ to $\mathcal{O}(1)$, as it no longer depends on the network depth $L$. We note that DKP-PC goes beyond the incremental predictive coding (iPC) algorithm (Salvatori et al., 2024), which delivers more stable training by alternating between neural activity updates Eq. (14) and feedforward weight updates Eq. (15). While iPC partially unlocks PC's parallelization potential, it requires full-batch training to do so, whereas DKP-PC achieves this independently of the training batch size.

## 4 RESULTS

In this section, we assess DKP-PC against BP, DKP, PC, iPC, and center-nudging PC (CN-PC), in terms of classification performance and training speed.

*Setup* – We evaluate the scalability of DKP-PC from multi-layer perceptrons (MLPs) to VGG-like convolutional neural networks (CNNs) (Simonyan & Zisserman, 2014). For the MLP experiments, a three-layer architecture is evaluated on MNIST and Fashion-MNIST (Yann, 2010; Xiao et al., 2017). For the CNN experiments, we assess the performance of VGG-7 and VGG-9 on the CIFAR-10, CIFAR-100, and Tiny ImageNet datasets (Krizhevsky et al., 2009; Le & Yang, 2015). For comparability with prior PC works and to facilitate future benchmarking, we employ the architectures reported by Pinchetti et al. (2024) in their discriminative mode experiments, and report their performance for PC, iPC and BP. Additional implementation details are reported in Appendix A.5. All implementations are based on the PyTorch framework and are available on GitHub. [2]

*Classification performance* – The classification results are summarized in Table 1. For the MLP architecture, all algorithms achieve comparable performance on both MNIST and FMNIST, with local algorithms even surpassing the test accuracy of BP. For VGG-7 and VGG-9 on CIFAR-10 and CIFAR-100, DKP-PC outperforms DKP, PC and iPC, achieving up to $14\%$ higher top-1 accuracy for VGG-9 on CIFAR-100 compared to standard PC, and $9\%$ higher top-1 accuracy than its more stable variant, iPC. However, CN-PC is the best local learning algorithm for all the settings mentioned so far. When moving to the Tiny ImageNet dataset, representing the most complex one evaluated in our experiments, we can see that DKP-PC outperforms all the local learning algorithms, achieving a final test accuracy of $35.04\%$, compared to $31.50\%$ for CN-PC. Furthermore, DKP-PC outperforms vanilla DKP in every setup evaluated, marking a gap of even $13\%$ top-1 accuracy for VGG-7 on CIFAR-100. Interestingly, by leveraging the complementary strengths of DKP and PC, DKP-PC consistently delivers higher accuracy than either method alone and substantially narrows the performance gap with BP, particularly in deeper architectures where local learning typically struggles.

*Training speed* – Table 2 shows the training times for one epoch, in seconds, for BP, DKP, PC, iPC, and DKP-PC, averaged over 5 trials, using the same experimental settings as in Table 1. CN-PC is omitted since, after nudging the final layer, its training dynamics match those of standard PC (up to the update sign (Pinchetti et al., 2024)). Importantly, DKP-PC requires only a single PC inference step to achieve the accuracies reported in Table 1. In contrast, PC models typically need a number of inference steps equal to or larger than the network depth to reach the reported accuracy (Pinchetti et al., 2024). Consequently, in our timing evaluation, we set the number of PC inference steps equal to the network depth. Therefore, the reported PC training times should be interpreted as a lower bound. All measurements were performed using PyTorch on an NVIDIA RTX A6000 GPU where the parallelization opportunities offered by DKP and DKP-PC have not been leveraged, as they would require the use of custom CUDA kernels[3]. These models were thus executed sequentially, a setting in which BP naturally exploits highly-optimized hardware mapping and execution. Therefore, despite only highlighting speedup through reduced inference-phase steps and not through parallelization, the speedup achieved by DKP-PC compared to other PC algorithms

---

[2]Link omitted to preserve the double-blind review process. The GitHub repository will also contain all hyperparameters for reproducibility.

[3]A standard parallelization of DKP-PC in PyTorch introduces significant thread management and synchronization overhead, which cancels out the potential speedup.

Table 1: Test accuracy in % (mean ± standard deviation) averaged over 5 random seeds. Results for PC, iPC, CN-PC and BP are taken from Pinchetti et al. (2024). The best results among local algorithms are highlighted in bold.

| % Accuracy | DKP | PC | iPC | CN-PC | **DKP-PC** | BP |
|---|---|---|---|---|---|---|
| **MLP** | | | | | | |
| MNIST | $98.03^{\pm 0.10}$ | $98.26^{\pm 0.04}$ | $\mathbf{98.45^{\pm 0.09}}$ | $98.23^{\pm 0.09}$ | $98.02^{\pm 0.09}$ | $98.29^{\pm 0.08}$ |
| FashionMNIST | $88.86^{\pm 0.13}$ | $89.58^{\pm 0.13}$ | $\mathbf{89.90^{\pm 0.06}}$ | $89.56^{\pm 0.05}$ | $89.42^{\pm 0.25}$ | $89.48^{\pm 0.07}$ |
| **VGG-7** | | | | | | |
| CIFAR-10 | $77.98^{\pm 0.39}$ | $81.91^{\pm 0.30}$ | $80.15^{\pm 0.18}$ | $\mathbf{88.40^{\pm 0.12}}$ | $82.36^{\pm 0.18}$ | $89.91^{\pm 0.10}$ |
| CIFAR-100 (Top-1) | $36.96^{\pm 0.62}$ | $37.52^{\pm 2.60}$ | $43.99^{\pm 0.30}$ | $\mathbf{64.76^{\pm 0.17}}$ | $50.42^{\pm 0.38}$ | $65.36^{\pm 0.15}$ |
| CIFAR-100 (Top-5) | $64.93^{\pm 0.46}$ | $66.73^{\pm 2.37}$ | $73.23^{\pm 0.30}$ | $\mathbf{84.65^{\pm 0.18}}$ | $77.24^{\pm 0.60}$ | $84.41^{\pm 0.26}$ |
| **VGG-9** | | | | | | |
| CIFAR-10 | $77.12^{\pm 0.33}$ | $75.33^{\pm 0.25}$ | $79.02^{\pm 0.21}$ | $\mathbf{87.19^{\pm 0.41}}$ | $81.95^{\pm 0.19}$ | $90.02^{\pm 0.18}$ |
| CIFAR-100 (Top-1) | $46.07^{\pm 1.00}$ | $39.57^{\pm 0.18}$ | $44.76^{\pm 0.40}$ | $\mathbf{58.92^{\pm 1.61}}$ | $53.80^{\pm 0.64}$ | $65.51^{\pm 0.23}$ |
| CIFAR-100 (Top-5) | $72.80^{\pm 1.06}$ | $66.90^{\pm 0.26}$ | $72.88^{\pm 0.29}$ | $\mathbf{81.56^{\pm 0.63}}$ | $79.26^{\pm 0.63}$ | $84.70^{\pm 0.28}$ |
| Tiny ImageNet (Top-1) | $29.61^{\pm 0.60}$ | $21.78^{\pm 0.15}$ | $26.34^{\pm 0.03}$ | $31.50^{\pm 0.70}$ | $\mathbf{35.04^{\pm 2.64}}$ | $65.51^{\pm 0.23}$ |
| Tiny ImageNet (Top-5) | $53.03^{\pm 0.73}$ | $44.43^{\pm 0.09}$ | $50.48^{\pm 0.05}$ | $54.67^{\pm 0.68}$ | $\mathbf{58.61^{\pm 3.12}}$ | $84.70^{\pm 0.28}$ |

Table 2: Training time per epoch in seconds (mean ± standard deviation), averaged over 5 trials.

| Seconds | DKP | PC | iPC | **DKP-PC** | BP |
|---|---|---|---|---|---|
| **MLP** | | | | | |
| MNIST | $4.70^{\pm 0.10}$ | $4.71^{\pm 0.06}$ | $4.79^{\pm 0.09}$ | $4.74^{\pm 0.09}$ | $4.70^{\pm 0.06}$ |
| FashionMNIST | $4.62^{\pm 0.06}$ | $4.77^{\pm 0.13}$ | $5.07^{\pm 0.13}$ | $4.70^{\pm 0.14}$ | $4.62^{\pm 0.07}$ |
| **VGG-7** | | | | | |
| CIFAR-10 | $7.21^{\pm 0.26}$ | $31.39^{\pm 0.20}$ | $54.48^{\pm 0.12}$ | $11.13^{\pm 0.09}$ | $7.27^{\pm 0.19}$ |
| CIFAR-100 | $7.11^{\pm 0.06}$ | $31.48^{\pm 0.17}$ | $54.69^{\pm 0.22}$ | $11.67^{\pm 0.04}$ | $7.15^{\pm 0.04}$ |
| **VGG-9** | | | | | |
| CIFAR-10 | $7.21^{\pm 0.09}$ | $34.10^{\pm 0.15}$ | $69.34^{\pm 0.10}$ | $12.06^{\pm 0.03}$ | $7.09^{\pm 0.08}$ |
| CIFAR-100 | $7.17^{\pm 0.06}$ | $34.18^{\pm 0.04}$ | $69.73^{\pm 0.09}$ | $12.53^{\pm 0.01}$ | $6.95^{\pm 0.05}$ |
| Tiny ImageNet | $35.37^{\pm 3.57}$ | $158.48^{\pm 1.59}$ | $303.14^{\pm 0.19}$ | $54.10^{\pm 0.13}$ | $38.27^{\pm 5.45}$ |

is notable. Indeed, while on very small networks GPU and kernel overheads dominate the runtime, making algorithmic differences negligible, as depth increases the computational gap grows sharply. On the evaluated CNNs, DKP-PC delivers approximately an average training time reduction of $64\%$ compared to PC and $81\%$ compared to iPC. We elaborate further on the computational trade-offs of DKP-PC in Appendix A.6.

## 5 CONCLUSION AND FUTURE WORK

We introduced DKP-PC, the first training algorithm that releases PC's feedback error delay and exponential decay toward enabling fully parallelized, local learning. We evaluated its classification performance, training speed, and computational efficiency against BP, DKP, PC, and iPC. Our results show that, by accelerating PC with DKP, DKP-PC scales better than the evaluated local-learning algorithms, while exhibiting a substantial improvement in computational efficiency and training time compared to PC and its newer variants iPC and CN-PC. These results indicate that local learning rules can approach BP's efficiency while narrowing the scalability gap, which is particularly relevant for neuromorphic computing and on-chip learning (Millidge et al., 2022a; Frenkel et al., 2023). Future work should focus on custom CUDA kernels to address the thread management and synchronization overheads of the current PyTorch implementation. Indeed, despite already a

significant speed-up compared to PC, the training time of DKP-PC will still lag behind that of BP as long as parallelization opportunities are not fully exploited. Furthermore, as feedback matrices introduce memory overhead, sparsity and quantization of feedback weights should be explored, as incentivized by prior work (Crafton et al., 2019; Han & Yoo, 2019). Lastly, this novel combination of feedback-alignment methods and PC might pave the way for a new class of algorithms focused on exploiting the synergy between the two frameworks and leveraging their specific dynamics. An interesting research direction is to directly use the feedback information to perturb the neural activity dynamics, without relying on a preliminary weight update step, thereby outlining faster and more efficient local update rules for the neural activity dynamics. Future work could also focus on a tailored integration of DKP with advanced PC variants, such as nudging PC based on equilibrium propagation (Scellier & Bengio, 2017; Scellier et al., 2023; Pinchetti et al., 2024), combining their dynamics with the DKP learning rules for both forward and feedback weights. This integration could allow the different formulations to complement each other and further reduce the performance gap with BP.

## 6 ETHICS STATEMENT

We affirm that this work fully adheres to the ICLR Code of Ethics. This study does not involve human subjects and relies exclusively on well-known open-source datasets. The research presents no conflicts of interest and does not raise any privacy, security, or safety concerns. All experiments, results, and comparisons have been conducted and presented with scientific integrity, ensuring accuracy, transparency, and reproducibility. Care has been taken to ensure fairness in the evaluation and comparison of methods, avoiding bias in reporting or interpretation.

## 7 REPRODUCIBILITY STATEMENT

We are fully committed to ensuring the reproducibility of our results and research. We provide detailed mathematical derivations and pseudocode to support a thorough theoretical understanding of our method. Additionally, the appendix contains a comprehensive description of the experimental settings and technical details to enable fair and complete reproducibility. Upon acceptance, we will include a GitHub repository containing all code, experiments, and parameters necessary to reproduce all results reported in this manuscript. These measures ensure transparency, integrity, and reproducibility in accordance with the ICLR Code of Ethics.

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

## A  APPENDIX

### A.1  CONVERGENCE OF FEEDBACK MATRICES UNDER THE DIRECT KOLEN–POLLACK ALGORITHM

In this appendix, we extend the empirical observations of Webster et al. (2020) by providing a mathematical argument for why DKP achieves a better alignment with BP than DFA. While their work demonstrates this empirically, it does not provide a formal theoretical justification, instead attributing the behavior to analogies with KP. However, while it has been proven for KP that the feedback matrices $\Psi_\ell$ converge to $\Theta_\ell^\top$ for all layers (Kolen & Pollack, 1994; Akrout et al., 2019), in DKP this result strictly holds only for the last layer, as all other hidden layers have a dimensionality mismatch between $\Psi_\ell$ and $\Theta_\ell$. Here, we offer a novel theoretical perspective on the work of Webster

et al. (2020), demonstrating that DKP drives the feedback matrices toward values approximating the chain of transposed forward weights, similar to BP, up to the Moore-Penrose pseudoinverse.

Let us consider a feedforward neural network with $L$ layers, where $d_\ell$ denotes the number of neurons at layer $\ell$. The weight matrix $\Theta_\ell \in \mathbb{R}^{d_{\ell+1} \times d_\ell}$ maps activations from layer $x_\ell \in \mathbb{R}^{d_\ell}$ to the next layer $x_{\ell+1} \in \mathbb{R}^{d_{\ell+1}}$, according to

$$x_{\ell+1} = f(\Theta_\ell x_\ell), \tag{21}$$

where $f(\cdot)$ denotes an arbitrary non-linear activation function. In the following derivations, we assume an identity activation function, so that the derivative of the activation function can be omitted.

In DFA, error feedback is provided by fixed random matrices $\Psi_\ell \in \mathbb{R}^{d_\ell \times d_L}$ that project the output error term $\delta_L$ directly to each hidden layer $\ell$. These matrices replace the layer-specific error term $\delta_\ell$ used in BP, computed by transporting the error backward from the next layer through the transposed weight matrix $\Theta_{\ell-1}^\top$, as described by

$$\delta_\ell = \Theta_{\ell+1}^\top \delta_{\ell+1}. \tag{22}$$

In DFA, as illustrated in Figure 1, this layer-specific error term $\delta_\ell$ is instead approximated by

$$\tilde{\delta}_\ell = \Psi_\ell \delta_L. \tag{23}$$

In contrast, DKP allows the feedback matrices to be updated following the local update rule

$$\Delta\Psi_\ell = x_\ell \delta_L^\top. \tag{24}$$

With both forward and feedback weights subject to a decay term, this learning has been empirically shown to enable $\Psi_\ell$ to provide a more BP-aligned update for $\Theta_\ell$ compared to standard DFA (Webster et al., 2020). We extend their work by demonstrating that the feedback matrices gradually align with the pseudoinverses of the forward weight matrices in a recursive dependency, thereby yielding a closer approximation of BP error propagation compared to DFA.

Starting from the last layer, the update rule for the weight matrix preceding it, denoted as $\Theta_{L-1} \in \mathbb{R}^{d_L \times d_{L-1}}$, is the same for BP, DFA, and DKP, and is given by

$$\Delta\Theta_{L-1} = -\alpha \left( \delta_L x_{L-1}^\top + \Theta_{L-1} \right), \tag{25}$$

where $\alpha \in (0, 1)$ is the learning rate, and the second term of the update is the weight decay term. According to Eq. (24) and assuming the same learning rate for the feedback weights, the update rule for the feedback matrix connecting the last layer to the penultimate one is given by

$$\Delta\Psi_{L-1} = -\alpha \left( x_{L-1} \delta_L^\top + \Psi_{L-1} \right). \tag{26}$$

In this specific case, $\Theta_{L-1}$ has the shape of $\Psi_{L-1}^\top$ and their updates are transposes of each other. As training progresses, both matrices converge to the same value, since the contribution of the initial condition vanishes under the effect of the learning rate $\alpha$ (Kolen & Pollack, 1994; Akrout et al., 2019). Indeed, following KP, by defining

$$\Omega_{L-1}(t+1) = \Theta_{L-1}(t+1) - \Psi_{L-1}^\top(t+1), \tag{27}$$

and using the update rules Eq. (25) and Eq. (26), we obtain

$$
\begin{aligned}
\Omega_{L-1}(t+1) &= (\Theta_{L-1}(t) + \Delta\Theta_{L-1}(t)) - \left( \Psi_{L-1}^\top(t) + \Delta\Psi_{L-1}^\top(t) \right) \\
&= \Theta_{L-1}(t) - \Psi_{L-1}^\top(t) - \alpha \left( \delta_L(t) x_{L-1}^\top(t) + \Theta_{L-1}(t) - \delta_L(t) x_{L-1}^\top(t) - \Psi_{L-1}^\top(t) \right) \\
&= \Theta_{L-1}(t) - \Psi_{L-1}^\top(t) - \alpha \left( \Theta_{L-1}(t) - \Psi_{L-1}^\top(t) \right) \\
&= (1 - \alpha) \left( \Theta_{L-1}(t) - \Psi_{L-1}^\top(t) \right) \\
&= (1 - \alpha) \Omega_{L-1}(t),
\end{aligned}
\tag{28}
$$

We can now further unroll Eq. (28) in time, resulting in

$$
\begin{aligned}
\Omega_{L-1}(t+1) &= (1 - \alpha)^t \Omega_{L-1}(0) \\
&= (1 - \alpha)^t \left( \Theta_{L-1}(0) - \Psi_{L-1}^\top(0) \right).
\end{aligned}
\tag{29}
$$

Therefore, $\Omega_{L-1}(t)$ converges to zero as training progresses, with the initial difference between the forward and feedback matrices decaying exponentially due to the learning rate $\alpha$. In other words, we have that

$$\lim_{t\to\infty} \Omega_{L-1}(t) = 0 \quad \implies \quad \lim_{t\to\infty} \Psi_{L-1}^\top(t) = \Theta_{L-1}(t). \tag{30}$$

The DKP update rule for $\Theta_{L-2}$ is given by

$$\Delta\Theta_{L-2} = -\alpha\left(\tilde{\delta}_{L-1}x_{L-2}^\top + \Theta_{L-2}\right)$$
$$= -\alpha\left(\Psi_{L-1}\delta_L x_{L-2}^\top + \Theta_{L-2}\right), \tag{31}$$

which effectively approximates the BP update for $\Theta_{L-2}$ and ultimately matches it as $t \to \infty$, as using Eq. (30) yields

$$\lim_{t\to\infty}\Delta\Theta_{L-2} = -\alpha\left(\Theta_{L-1}^\top\delta_L x_{L-2}^\top + \Theta_{L-2}\right). \tag{32}$$

Here and throughout the rest of this appendix, we omit the time index since we consider the limit $t \to \infty$. Therefore, we approximate $\Psi_{L-1}$ by $\Theta_{L-1}^\top$, noting that this approximation introduces a small error, since exact equality holds only at the limit.

Unfortunately, the convergence obtained for $\Psi_{L-1}$ in Eq. (30) does not directly apply to $\Psi_{L-2} \in \mathbb{R}^{d_{L-2}\times d_L}$, as its dimensions do not match those of $\Theta_{L-2} \in \mathbb{R}^{d_{L-1}\times d_{L-2}}$. The update rule for $\Psi_{L-2}$, given by

$$\Delta\Psi_{L-2} = -\alpha\left(x_{L-2}\delta_L^\top + \Psi_{L_2}\right), \tag{33}$$

can be substituted into Eq. (32), leading to

$$\Delta\Theta_{L-2} = -\alpha\left(\Theta_{L-1}^\top\delta_L x_{L-2}^\top + \Theta_{L-2}\right)$$
$$= -\alpha\left(\Theta_{L-1}^\top\left(-\alpha^{-1}\Delta\Psi_{L-2}^\top - \Psi_{L-2}^\top\right) + \Theta_{L-2}\right) \tag{34}$$
$$= \Theta_{L-1}^\top\Delta\Psi_{L-2}^\top - \alpha\left(\Theta_{L-2} - \Theta_{L-1}^\top\Psi_{L-2}^\top\right).$$

Here, we neglect the decay terms, since they vanish asymptotically during training, as previously discussed for $\Theta_{L-1}$ and $\Psi_{L-1}$. We can now transpose both sides, and multiply them by $\Theta_{L-1}^\top$, resulting in

$$\Delta\Psi_{L-2}\Theta_{L-1}\Theta_{L-1}^\top = \Delta\Theta_{L-2}^\top\Theta_{L-1}^\top. \tag{35}$$

This allows us to link the update of $\Psi_{L-2}$ to that of $\Theta_{L-2}$ through

$$\Delta\Psi_{L-2} = \Delta\Theta_{L-2}^\top\Theta_{L-1}^\top\left(\Theta_{L-1}\Theta_{L-1}^\top\right)^{-1}$$
$$= \Delta\Theta_{L-2}^\top\Theta_{L-1}^+, \tag{36}$$

where $\Theta_{L-1}^+ \in \mathbb{R}^{d_{L-1}\times d_L}$ is the Moore-Penrose pseudoinverse of $\Theta_{L-1}$, assuming the latter has full row rank. On the one hand, the BP update of $\Theta_{L-3}$ is given by

$$\Delta\Theta_{L-3} = -\alpha\left(\delta_{L-2}x_{L-3}^\top + \Theta_{L-3}\right)$$
$$= -\alpha\left(\Theta_{L-2}^\top\delta_{L-1}x_{L-3}^\top + \Theta_{L-3}\right) \tag{37}$$
$$= -\alpha\left(\Theta_{L-2}^\top\Theta_{L-1}^\top\delta_L x_{L-3}^\top + \Theta_{L-3}\right).$$

On the other hand, by making use of Eq. (36), which is valid under the previously mentioned assumptions and approximations, the DKP update can be expressed as

$$\Delta\Theta_{L-3} = -\alpha\left(\tilde{\delta}_{L-2}x_{L-3}^\top + \Theta_{L-3}\right)$$
$$= -\alpha\left(\Psi_{L-2}^\top\delta_L x_{L-3}^\top + \Theta_{L-3}\right) \tag{38}$$
$$= -\alpha\left(\Theta_{L-2}^\top\Theta_{L-1}^+\delta_L x_{L-3}^\top + \Theta_{L-3}\right).$$

Hence, the DKP update of $\Theta_{L-3}$ approximates the BP one, and is exact if $\Theta_{L-1}^+ = \Theta_{L-1}^\top$, meaning that $\Theta_{L-1}$ is orthogonal. We now move on to the DKP update of $\Psi_{L-3}$, given by

$$\Delta\Psi_{L-3} = -\alpha\left(x_{L-3}\delta_L^\top + \Psi_{L-3}\right). \tag{39}$$

By repeating the same procedure as for $\Psi_{L-2}$, we substitute Eq. (39) into Eq. (38), resulting in

$$\Delta\Theta_{L-3} = -\alpha\left(\Theta_{L-2}^\top\Theta_{L-1}^+\left(-\alpha^{-1}\Delta\Psi_{L-3} - \Psi_{L-3}\right)^\top + \Theta_{L-3}\right)$$
$$= \Theta_{L-2}^\top\Theta_{L-1}^+\Delta\Psi_{L-3}^\top - \alpha\left(\Theta_{L-3} - \Theta_{L-2}^\top\Theta_{L-1}^+\Psi_{L-3}^\top\right). \tag{40}$$

We now again do not consider the terms decaying with time, as they tend to zero as the training goes on, and focus only on the term the update converges to. As done previously, after dropping the decay term, we transpose both sides, yielding

$$\Delta\Psi_{L-3}\left(\left(\Theta_{L-1}^+\right)^\top\Theta_{L-2}\right) = \Delta\Theta_{L-3}^\top, \tag{41}$$

and multiply them by $\left(\left(\Theta_{L-1}^+\right)^\top\Theta_{L-2}\right)^\top$, leading to

$$\Delta\Psi_{L-3}\left(\left(\Theta_{L-1}^+\right)^\top\Theta_{L-2}\right)\left(\left(\Theta_{L-1}^+\right)^\top\Theta_{L-2}\right)^\top = \Delta\Theta_{L-3}^\top\left(\left(\Theta_{L-1}^+\right)^\top\Theta_{L-2}\right)^\top. \tag{42}$$

Lastly, by multiplying both sides by the inverse of the product of matrices on the right of $\Delta\Psi_{L-3}$, we obtain

$$\begin{aligned}
\Delta\Psi_{L-3} &= \Delta\Theta_{L-3}^\top\left(\left(\Theta_{L-1}^+\right)^\top\Theta_{L-2}\right)^\top\left[\left(\left(\Theta_{L-1}^+\right)^\top\Theta_{L-2}\right)\left(\left(\Theta_{L-1}^+\right)^\top\Theta_{L-2}\right)^\top\right]^{-1} \\
&= \Delta\Theta_{L-3}^\top\left(\left(\Theta_{L-1}^+\right)^\top\Theta_{L-2}\right)^+, \\
&= \Delta\Theta_{L-3}^\top\left(\left(\Theta_{L-1}^\top\right)^+\Theta_{L-2}\right)^+,
\end{aligned} \tag{43}$$

where again, the feedback matrix includes a chain of forward matrix pseudoinverses. More generally, under the assumption of $t\to\infty$ and that $\Theta_\ell$ is a full row rank rectangular matrix, we have

$$\Psi_{L-\ell} = \begin{cases} \Theta_{L-\ell}^\top & \text{if } \ell = 1, \\ \Theta_{L-\ell}^\top\left(\Psi_{L-\ell+1}^\top\right)^+ & \text{if } 1 < \ell < L, \end{cases} \tag{44}$$

It should be noted that in practice, the assumptions we made are never perfectly met, since neural network training does not proceed for an infinite number of iterations, meaning that the decay terms we neglected do not completely vanish, and also involves the derivatives of the non-linear activation functions. Nonetheless, our derivation demonstrates how DKP extends DFA by updating the feedback random matrices with terms that also appear in BP's error propagation, providing a clearer understanding of why DKP achieves better alignment and consequently improves performance compared to standard DFA Webster et al. (2020).

## A.2 ERROR PROPAGATION DELAY

In this section, we introduce and provide a formal proof of Theorem 1, which quantifies the delay in error propagation in forward-initialized PC networks. Specifically, we show that the feedback error signal reaches a given layer with a delay equal to its distance from the output (Zahid et al., 2023).

**Theorem 1** (Error propagation delay). *Consider a forward-initialized PC network with discrete-time updates. Assuming an incorrect prediction, the neural activity $\phi_\ell$ at layer $\ell$ requires at least $\hat{t} = L - \ell$ inference-phase steps before it deviates from equilibrium and begins to evolve according to Eq. (14) (proof provided in Appendix A.2).*

*Proof.* Let us consider a forward-initialized PC network with $L+1$ layers, where the neural activity evolves in discrete time steps $t \in \mathbb{N}_0$, and each layer is initialized as $\phi_\ell(0) = f(\Theta_{\ell-1}\phi_{\ell-1}(0))$. The error neurons at layer $\ell$ are defined as $\epsilon_\ell(0) = \phi_\ell(0) - f(\Theta_{\ell-1}\phi_{\ell-1}(0))$. By construction, after forward initialization, all error neurons vanish and the network's dynamics is at equilibrium:

$$\begin{aligned}
\|\epsilon_\ell(0)\|_2^2 &= \|\phi_\ell(0) - f(\Theta_{\ell-1}\phi_{\ell-1}(0))\|_2^2 \\
&= \|\phi_\ell(0) - \phi_\ell(0)\|_2^2 \\
&= 0 \quad \text{for} \quad 0 < \ell \leq L.
\end{aligned} \tag{45}$$

At the beginning of the inference phase, where the network's neural activity evolves to minimize Eq. (12), we clamp the target vector $y$ to the output layer $\phi_L$. Assuming an incorrect prediction, i.e., $y - f(\Theta_{L-1}\phi_{L-1}(0)) \neq 0$, the prediction error satisfies

$$\|\epsilon_L(0)\|_2^2 = \|y - f(\Theta_{L-1}\phi_{L-1}(0))\|_2^2 \geq 0. \tag{46}$$

Consequently, according to Eq. (14), the activity at layer $L - 1$ has a non-zero update at $t = 1$:

$$\phi_{L-1}(1) = \phi_{L-1}(0) + \Delta\phi_\ell(0)$$

$$= \phi_{L-1}(0) - \gamma\frac{\partial F}{\partial \phi_{L-1}}(0)$$

$$= \phi_{L-1}(0) - \gamma\Big(J_{\phi_{L-1}}(0)^\top \epsilon_L - \epsilon_{L-1}\Big) \quad (47)$$

$$= \phi_{L-1}(0) - \gamma\Big(J_{\phi_{L-1}}(0)^\top \epsilon_L\Big),$$

where $J_{\phi_\ell}(0) = \frac{\partial f(\Theta_\ell \phi_\ell)}{\partial \phi_\ell}(0) \in \mathbb{R}^{d_{\ell+1} \times d_\ell}$ is the Jacobian matrix of prediction by layer $\ell$ with respect to its neural activity. For all preceding layers, we have that $\Delta\phi_\ell(0) = 0$, since both $\epsilon_\ell$ and $\epsilon_{\ell+1}$ are null.

After $\phi_{L-1}$ has been updated at time $t = 1$, the corresponding error becomes non-zero:

$$\|\epsilon_{L-1}(1)\|_2^2 = \|\phi_{L-1}(1) - f(\Theta_{L-2}, \phi_{L-2}(1))\|_2^2$$

$$= \|\phi_{L-1}(1) - f(\Theta_{L-2}, \phi_{L-2}(0))\|_2^2$$

$$= \|\phi_{L-1}(1) - \phi_{L-1}(0)\|_2^2 \quad (48)$$

$$= \|\Delta\phi_{L-1}(1)\|_2^2$$

$$\geq 0.$$

At the subsequent timestamp $t = 2$, layer $L - 2$ also receives a non-zero error and updates accordingly:

$$\phi_{L-2}(2) = \phi_{L-2}(1) + \Delta\phi_{L-2}(1)$$

$$= \phi_{L-2}(1) - \gamma\frac{\partial F}{\partial \phi_{L-2}}(1)$$

$$= \phi_{L-2}(0) - \gamma\frac{\partial F}{\partial \phi_{L-2}}(0) \quad (49)$$

$$= \phi_{L-2}(0) - \gamma\Big(J_{\phi_{L-2}}(0)^\top \epsilon_{L-1} - \epsilon_{L-2}\Big)$$

$$= \phi_{L-2}(0) - \gamma\Big(J_{\phi_{L-2}}(0)^\top \epsilon_{L-1}\Big),$$

where $\phi_{L-2}(1) = \phi_{L-2}(0)$ and $\frac{\partial F}{\partial \phi_{L-2}}(1) = \frac{\partial F}{\partial \phi_{L-2}}(0)$, since $\phi_{L-2}$ has remained unchanged at $t = 1$, due to $\epsilon_{L-2}(0) = \epsilon_{L-1}(0) = 0$. Again, at time $t = 2$ all previous layers $\phi_\ell(2)$, $0 < \ell < L - 2$ remain unchanged, as $\epsilon_{L-\ell}(1) = \epsilon_{L-\ell+1}(1) = 0$. By induction, it follows that under these assumptions, any layer $\ell$ requires at least $\hat{t} = L - \ell$ timestamps to update its neural activity and corresponding error neurons.

For the general case, at a specific time $t$, the error neurons at layer $L - t$ can be expressed as

$$\|\epsilon_{L-t}(t)\|_2^2 = \|\phi_{L-t}(t) - f(\Theta_{L-t-1}, \phi_{L-t-1}(t - 1))\|_2^2$$

$$= \|\phi_{L-t}(t) - f(\Theta_{L-t-1}, \phi_{L-t-1}(0))\|_2^2 \quad (50)$$

$$= \|\phi_{L-t}(t) - \phi_{L-t}(0)\|_2^2,$$

where $\phi_{L-t}(t) - \phi_{L-t}(0) \neq 0$ if and only if $\phi_{L-t}(t) \neq \phi_{L-t}(0)$. Crucially, this condition can occur only after $\hat{t} = L - \ell$ timestamps. According to Eq. (14), the update of layer $L - t$ depends on the errors from the current and subsequent layers, $\epsilon_{L-t}$ and $\epsilon_{L-t+1}$, respectively. Since $\epsilon_{L-t}$ is itself blocked until $\phi_{L-t}$ changes, the driving term is provided by $\epsilon_{L-t+1}$. However, the latter becomes non-zero only after the previous layer has been updated. Therefore, the propagation of activity changes and error signals strictly follows the network's hierarchy, advancing at most one layer per timestamp, starting from $\epsilon_L$ at $t = 0$ and reaching layer $\ell$ only after $\hat{t} = L - \ell$ steps.

### A.3 ERROR EXPONENTIAL DECAY

In this section, we introduce and provide a formal proof of Theorem 2, showing that the squared-$\ell_2$-norm of the feedback error signal for a layer $\ell$ generated at time $\hat{t} = L - \ell$ is bounded by a quantity that decays exponentially proportional to $t$ (Goemaere et al., 2025).

**Theorem 2** (Error exponential decay). *Consider a forward-initialized PC network with discrete-time updates. Assuming an incorrect prediction, the squared $\ell_2$-norm of the feedback error signal at layer $\ell$, $\|\epsilon_\ell(\hat{t})\|_2^2$, at time $\hat{t} = L - \ell$, is upper-bounded by a quantity that decays $\propto \gamma^{2(L-\ell)}$ (proof provided in Appendix A.3).*

*Proof.* Let us consider a forward-initialized PC network with $L + 1$ layers, where the neural activity evolves in discrete time steps $t \in \mathbb{N}_0$, and each hidden layer $\phi_\ell(t)$ is initialized as $\phi_\ell(0) = f(\Theta_{\ell-1}\phi_{\ell-1}(0))$. The error neurons are defined as $\epsilon_\ell(t) = \phi_\ell(t) - f(\Theta_{\ell-1}\phi_{\ell-1}(t))$. Here, we make the time dependence of neural activities explicit, as they evolve with time during the inference phase of PC. Considering the update of an arbitrary neuron $\phi_{\ell-1}$, $1 < \ell \leq L$ at time $\hat{t} + 1$, with $\hat{t} = L - \ell$:

$$\phi_{\ell-1}(\hat{t} + 1) = \phi_{\ell-1}(\hat{t}) - \gamma \frac{\partial F}{\partial \phi_{\ell-1}}(\hat{t}), \tag{51}$$

where we can move the on the left side the current neural activity value, obtaining:

$$\phi_{\ell-1}(\hat{t} + 1) - \phi_{\ell-1}(\hat{t}) = -\gamma \frac{\partial F}{\partial \phi_{\ell-1}}(\hat{t})$$

$$\Delta \phi_{\ell-1}(\hat{t} + 1) = -\gamma \frac{\partial F}{\partial \phi_{\ell-1}}(\hat{t})$$

$$= -\gamma \left( \epsilon_{\ell-1}(\hat{t}) - \frac{\partial f\left(\Theta_{\ell-1}\phi_{\ell-1}(\hat{t})\right)}{\partial \phi_{\ell-1}(\hat{t})}^\top \epsilon_\ell(\hat{t}) \right) \tag{52}$$

$$= \gamma \left( \frac{\partial f\left(\Theta_{\ell-1}\phi_{\ell-1}(\hat{t})\right)}{\partial \phi_{\ell-1}(\hat{t})}^\top \epsilon_\ell(\hat{t}) \right),$$

where $\epsilon_{\ell-1}(\hat{t}) = 0$ is implied by Theorem 1, since $L - \ell - 1 < \hat{t}$. We defined $\frac{\partial f\left(\Theta_{\ell-1}\phi_{\ell-1}(\hat{t})\right)}{\partial \phi_{\ell-1}(\hat{t})} = J_{\phi_{\ell-1}} \in \mathbb{R}^{d_\ell \times d_{\ell-1}}$ the Jacobian matrix of $f$ at layer $\ell - 1$. Here, we have omitted time in $\phi_{\ell-1}$ for brevity, as $\phi_{\ell-1}(\hat{t}) = \phi_{\ell-1}(0)$, following again from Theorem 1. Continuing from Eq. (52), we expand the error neurons according to their definition and apply Theorem 1, resulting in:

$$\Delta \phi_{\ell-1}(\hat{t} + 1) = \gamma J_{\phi_{\ell-1}}^\top \left[ \phi_\ell(\hat{t}) - f\left(\Theta_{\ell-1}\phi_{\ell-1}(\hat{t})\right) \right]$$

$$= \gamma J_{\phi_{\ell-1}}^\top \left[ \phi_\ell(\hat{t} - 1) - \gamma \frac{\partial F}{\partial \phi_\ell}(\hat{t} - 1) - f\left(\Theta_{\ell-1}\phi_{\ell-1}(0)\right) \right]$$

$$= \gamma J_{\phi_{\ell-1}}^\top \left[ \phi_\ell(0) - \gamma \frac{\partial F}{\partial \phi_\ell}(\hat{t} - 1) - f(\Theta_{\ell-1}\phi_{\ell-1}(0)) \right] \tag{53}$$

$$= \gamma J_{\phi_{\ell-1}}^\top \left[ f\left(\Theta_{\ell-1}\phi_{\ell-1}(0)\right) - \gamma \frac{\partial F}{\partial \phi_\ell}(\hat{t} - 1) - f\left(\Theta_{\ell-1}\phi_{\ell-1}(0)\right) \right],$$

where on the first line we have substituted $\phi_\ell(\hat{t})$ with its definition in Eq. (51). According to Theorem 1, $\phi_\ell(\hat{t} - 1) = \phi_\ell(0)$ as $L - \ell < \hat{t} - 1$, thus this allows to re-write it using the forward initialization definition, as done on the fourth line. By repeating the same steps as in Eq. (52), we obtain

$$\Delta \phi_{\ell-1}(\hat{t} + 1) = \gamma J_{\phi_{\ell-1}}^\top \left[ -\gamma \frac{\partial F}{\partial \phi_\ell}(\hat{t} - 1) \right]$$

$$= \gamma J_{\phi_{\ell-1}}^\top \left[ -\gamma \left( \epsilon_\ell(\hat{t} - 1) - J_{\phi_\ell}^\top \epsilon_{\ell+1}(\hat{t} - 1) \right) \right] \tag{54}$$

$$= \gamma^2 J_{\phi_{\ell-1}}^\top J_{\phi_\ell}^\top \epsilon_{\ell+1}(\hat{t} - 1).$$

By unrolling the formulation backward until the last layer, we get:

$$\Delta \phi_{\ell-1}(\hat{t} + 1) = \gamma^{L-\ell+1} \left( \prod_{i=0}^{L-\ell} J_{\phi_{\ell-1+i}}^\top \right) \epsilon_L(0). \tag{55}$$

By substituting Eq. (52) in Eq. (55), we can continue as follows:

$$\gamma \left( J_{\phi_{\ell-1}}^{\top} \epsilon_\ell(\hat{t}) \right) = \gamma^{L-\ell+1} \left( \prod_{i=0}^{L-\ell} J_{\phi_{\ell-1+i}}^{\top} \right) \epsilon_L(0), \tag{56}$$

which can be further simplified by eliding common terms, resulting in:

$$\epsilon_\ell(\hat{t}) = \gamma^{L-\ell} \left( \prod_{i=1}^{L-\ell} J_{\phi_{\ell-1+i}}^{\top} \right) \epsilon_L(0). \tag{57}$$

By writing the squared-$\ell_2$ norm of Eq. (57), we finally derive the upper-bound for the error term:

$$\begin{aligned}
\|\epsilon_\ell(\hat{t})\|_2^2 &= \|\gamma^{L-\ell} \left( \prod_{i=1}^{L-\ell} J_{\phi_{\ell-1+i}}^{\top} \right) \epsilon_L(0)\|_2^2 \\
&\leq \gamma^{2(L-\ell)} \left( \prod_{i=1}^{L-\ell} \|J_{\phi_{\ell-1+i}}^{\top}\|_2^2 \right) \|\epsilon_L(0)\|_2^2.
\end{aligned} \tag{58}$$

### A.4 THEORETICAL AND EMPIRICAL ANALYSIS OF DKP-PC INTEGRATION

In this appendix, we further analyse DKP-PC, providing the reader with additional theoretical and empirical insights into the mechanisms through which the DKP and PC stages interact. Building on the results presented in the main text, we show how the single-step neural activity optimization introduced by the PC stage improves both the updates of the forward and feedback weights of the network.

### A.4.1 THEORETICAL ANALYSIS OF DKP-PC UPDATES

In this subsection, we derive analytically the neural activity and feedback weight updates within the proposed DKP-PC algorithm. We will follow the steps of the DKP-PC algorithm outlined in Algorithm 1. Besides, for the sake of mathematical tractability, we make the same assumption of linearity of the network as in Appendix A.1.

*1) Direct feedback alignment update* – First, the forward weights $\Theta_\ell$ are updated using the approximate gradients $\tilde{\delta}_\ell = \Psi_\ell \delta_L$, provided through the DKP feedback matrices $\Psi_\ell$. This results in updated forward weights

$$\tilde{\Theta}_\ell = \Theta_\ell + \Delta\Theta_\ell, \tag{59}$$

where the DKP forward weights' update is given by

$$\Delta\Theta_\ell = -\alpha\tilde{\delta}_{\ell+1}\phi_\ell^{\top} = -\alpha\Psi_{\ell+1}\delta_L\phi_\ell^{\top}. \tag{60}$$

*2) Inference phase* – The DKP forward weights' update is followed by a single update of the neural activity $\phi_\ell$, aiming to minimize the network's FE, as opposed to several steps for usual PC algorithms for spreading the error information. This leads to the following optimized neural activity:

$$\phi_\ell^* = \phi_\ell + \Delta\tilde{\phi}_\ell. \tag{61}$$

The updated neural activity $\phi_\ell^*$ now incorporates the information injected in the forward weights through the DKP update, as the error neurons are computed with the new forward weights values as $\tilde{\epsilon}_{\ell+1} = \phi_{\ell+1} - \tilde{\Theta}_\ell\phi_\ell$, as shown by computing the neural activity update $\Delta\tilde{\phi}_\ell$ based on the FE after

the DKP weight update $\tilde{F}$:

$$
\begin{aligned}
\Delta\tilde{\phi}_\ell &= -\gamma\frac{\partial\tilde{F}}{\partial\phi_\ell} \\
&= \gamma\left(\tilde{\Theta}_\ell^\top\tilde{\epsilon}_{\ell+1} - \tilde{\epsilon}_\ell\right) \\
&= \gamma\left[\tilde{\Theta}_\ell^\top\left(\phi_{\ell+1} - \tilde{\Theta}_\ell\phi_\ell\right) - \left(\phi_\ell - \tilde{\Theta}_{\ell-1}\phi_{\ell-1}\right)\right] \\
&= \gamma\left[(\Theta_\ell + \Delta\Theta_\ell)^\top\left(\phi_{\ell+1} - (\Theta_\ell + \Delta\Theta_\ell)\phi_\ell\right) - (\phi_\ell - (\Theta_{\ell-1} + \Delta\Theta_{\ell-1})\phi_{\ell-1})\right] \\
&= \gamma\left[(\Theta_\ell + \Delta\Theta_\ell)^\top\left(\phi_{\ell+1} - \Theta_\ell\phi_\ell - \Delta\Theta_\ell\phi_\ell\right) - (\phi_\ell - \Theta_{\ell-1}\phi_{\ell-1} - \Delta\Theta_{\ell-1}\phi_{\ell-1})\right] \\
&= \gamma\left[(\Theta_\ell + \Delta\Theta_\ell)^\top\left(\epsilon_{\ell+1} - \Delta\Theta_\ell\phi_\ell\right) - (\epsilon_\ell - \Delta\Theta_{\ell-1}\phi_{\ell-1})\right] \\
&= \gamma\left[\Theta_\ell^\top\epsilon_{\ell+1} - \epsilon_\ell + \Delta\Theta_\ell^\top\epsilon_{\ell+1} - (\Theta_\ell + \Delta\Theta_\ell)^\top\Delta\Theta_\ell\phi_\ell + \Delta\Theta_{\ell-1}\phi_{\ell-1}\right] \\
&= \gamma\left(\nabla_{\phi_\ell}F + \Delta\Theta_\ell^\top\epsilon_{\ell+1} - \tilde{\Theta}_\ell^\top\Delta\Theta_\ell\phi_\ell + \Delta\Theta_{\ell-1}\phi_{\ell-1}\right).
\end{aligned}
\tag{62}
$$

From Eq. (62), we can see how Eq. (61) can be reformulated as the original FE gradient with respect to the neural activity before the DKP weight update, denoted as $\nabla_{\phi_\ell}F$, plus terms resulting from the DKP weight update itself.

Note that, because of the forward-initialization of the network, we have $\phi_{\ell+1} = \Theta_\ell\phi_\ell$ for $\ell < L-1$, except at layer $L-1$, as $\phi_L$ has been clamped to the target label at the beginning of this stage. This implies that, for intermediate layers $\ell = 1$ to $L-1$, the error nodes $\epsilon_\ell$ are equal to zero for a single inference step. The DKP-PC neural activity update thus simplifies to

$$
\Delta\tilde{\phi}_\ell = \gamma\left(-\tilde{\Theta}_\ell^\top\Delta\Theta_\ell\phi_\ell + \Delta\Theta_{\ell-1}\phi_{\ell-1}\right).
\tag{63}
$$

By now substituting the weight update term from Eq. (60) in Eq. (63), we obtain

$$
\begin{aligned}
\Delta\tilde{\phi}_\ell &= \gamma\left(-\Theta_\ell^\top\Delta\Theta_\ell\phi_\ell - \Delta\Theta_\ell^\top\Delta\Theta_\ell\phi_\ell + \Delta\Theta_{\ell-1}\phi_{\ell-1}\right) \\
&= \alpha\gamma\left(\Theta_\ell^\top\Psi_{\ell+1}\delta_L\phi_\ell^\top\phi_\ell - \alpha\phi_\ell\delta_L^\top\Psi_{\ell+1}^\top\Psi_{\ell+1}\delta_L\phi_\ell^\top\phi_\ell - \Psi_\ell\delta_L\phi_{\ell-1}^\top\phi_{\ell-1}\right) \\
&= \alpha\gamma\left(\|\phi_\ell\|_2^2\Theta_\ell^\top\Psi_{\ell+1}\delta_L - \alpha\|\phi_\ell\|_2^2\|\Psi_{\ell+1}\delta_L\|_2^2\phi_\ell - \|\phi_{\ell-1}\|_2^2\Psi_\ell\delta_L\right) \\
&= \underbrace{\alpha\gamma\left(\|\phi_\ell\|_2^2\Theta_\ell^\top\tilde{\delta}_{\ell+1} - \|\phi_{\ell-1}\|_2^2\tilde{\delta}_\ell\right)}_{\text{Alignment term}} - \underbrace{\alpha^2\gamma\left(\|\phi_\ell\|_2^2\|\tilde{\delta}_{\ell+1}\|_2^2\phi_\ell\right)}_{\text{Regularization term}}.
\end{aligned}
\tag{64}
$$

The first term in the neural activity update quantifies the discrepancy between two estimates of the error at layer $\ell$, namely the direct projection of the output error $\delta_L$ onto layer $\ell$, given by $\Psi_\ell\delta_L = \tilde{\delta}\ell$, and the backward projection produced by PC from layer $\ell+1$ through the error node, given by $\Theta\ell^\top\Psi_{\ell+1}\delta_L = \Theta_\ell^\top\tilde{\delta}_{\ell+1}$. Note that these projections are respectively weighted by the squared $\ell_2$ norm of neural activities $\phi_{\ell-1}$ and $\phi_\ell$. The second term of the update is an activity regularization term whose strength evolves dynamically throughout training and proportionally to $\|\phi_\ell\|^2$ and $\|\tilde{\delta}_{\ell+1}\|^2 = \|\Psi_{\ell+1}\delta_L\|^2$. Note that this specific interpretation holds only in the single-step inference phase regime, which is anyway the one considered in this paper.

*3) Learning phase* – Following the consecutive updates of the forward weights $\Theta_\ell$ and the neural activity $\phi_\ell$, the PC forward weights' update, which aims to minimize the FE, is given by

$$
\begin{aligned}
\Delta\tilde{\Theta}_\ell &\propto \frac{\partial\tilde{F}}{\partial\tilde{\Theta}_\ell} \\
&\propto \tilde{\epsilon}_{\ell+1}\phi_\ell^{*\top}.
\end{aligned}
\tag{65}
$$

While the unrolled mathematical expression of this update under the DKP regime results in a complex formulation that offers limited intuition, we empirically demonstrate in Appendix A.4.2 that the alignment term injected into the neural activity, and consequently into the subsequent weight update, is essential for achieving a more stable and stronger alignment between feedback and forward weights, thereby providing clearer insight into its role within DKP-PC.

*4) Direct Kolen-Pollack update* – Finally, the feedback matrices $\Psi_\ell$ are updated following

$$\Delta \Psi_\ell \propto \phi_\ell^* \tilde{\epsilon}_L^\top. \tag{66}$$

As for the previous case, although the explicit derivation of this update leads to a complex expression even in the linear case, our experiments in Appendix A.4.2 again confirm that the alignment component introduced through the neural activity update, and thus propagated into the feedback matrices, is crucial for obtaining better alignment between feedback and forward weights, demonstrating its contribution within DKP-PC.

Our analysis shows that the initial DKP forward-weight update induces non-zero errors $\epsilon_\ell$ across all layers, enabling the neural activities to be updated in a single inference step. This feature solves two key limitations of PC: the error delay by generating non-zero errors right away, and the error exponential decay, by directly projecting the output error $\delta_L$ to each layer through the feedback weights. Furthermore, the optimized neural activity $\phi_\ell^*$ from DKP-PC includes alignment information that is successively injected into the PC forward weights' update and the DKP feedback weights' update. This leads to faster convergence than PC or DKP individually, as it enables earlier and stronger error signals and improves the alignment between forward and feedback pathways, producing weight updates that more closely approximate those of BP.

### A.4.2 Empirical Analysis of Gradient Alignment

In this subsection, we empirically analyse the alignment between the forward weight gradients computed according to DKP-PC and standard DKP algorithms, in comparison with BP. The alignment is quantified using the cosine similarity $\cos(\theta) \in [-1, 1]$, which measures the directional agreement between two arbitrary vectors $u \in \mathbb{R}^n$ and $v \in \mathbb{R}^n$ as follows:

$$\cos(\theta) = \frac{u \cdot v}{\|u\| \|v\|}, \tag{67}$$

where $\| \cdot \|$ denote the Euclidean norm. The cosine similarity reaches a maximum value of 1 when the vectors are perfectly aligned ($\theta = 0$), a minimum value of $-1$ when they are diametrically opposed ($\theta = \pi$), and a value of 0 when the vectors are orthogonal ($\theta = \pi/2$), indicating no directional correlation. Figure 3 depicts the alignment across the nine layers of the VGG9-like CNN from Section 4 trained for 50 epochs on CIFAR-100. For both DKP and DKP-PC networks, the hyperparameters correspond to the best-performing configuration reported in Appendix A.5. All curves consider instantaneous gradients, excluding weight decay and momentum contributions, and are smoothed using an exponential moving average with a window of 100 batches. The gradient for DKP-PC is computed by summing the gradients resulting from its DKP and PC phases. First, we analyse the alignment's behaviour of DKP-PC to that of standard DKP. Then, we analyse the effect of disabling the update of feedback and forward weights after computing the neural activity optimization, to empirically support our claims in the previous subsection.

From Figure 3, we can observe that, although standard DKP (brown curves) exhibits a positive gradient alignment with BP, its convergence is slow, and the alignment progressively deteriorates with increasing distance from the output layer, consistent with the analysis in Appendix A.1. Several batches are required before reaching high cosine similarity, and alignment degradation is also observed at the end of training, particularly in layers 6 and 7, consistently with observations made in the FA literature (Refinetti et al., 2021). In contrast, DKP-PC (yellow curves) achieves a faster and higher alignment with BP than standard DKP across all nine layers. DKP-PC indeed converges with fewer batches and exhibits better and more stable alignment throughout training. These observations not only align with the improved performance observed in the classification experiments but also support the theoretical analysis presented in the previous subsection. As shown in Eq. (64), the neural activity update incorporates both an alignment and a regularization terms, which contribute to the forward weight update in Eq. (65). Consequently, this stage can be interpreted as a regularization factor applied to the preliminary DKP update in DKP-PC, which over time stabilizes alignment and compensates for the error introduced by the Moore–Penrose pseudoinverse (for both forward and feedback weights). To further support this claim, Figure 3 also includes two versions of DKP-PC in which the PC forward weight update and the DKP feedback weight update have respectively been ablated. For DKP-PC without PC forward weight update (light blue curves), alignment collapses in all hidden layers and deteriorates even in the output layer, highlighting the key role of injecting

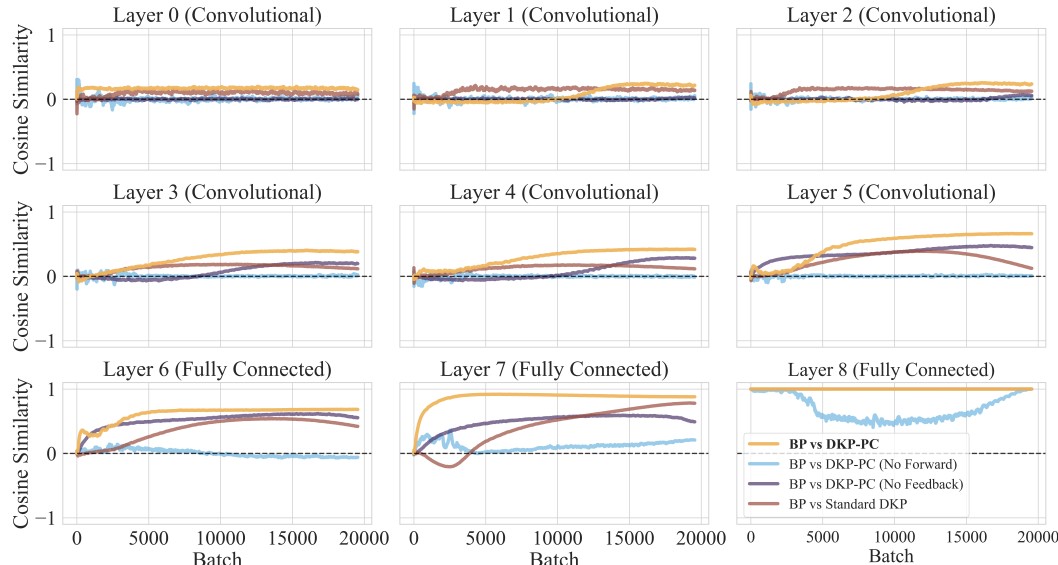

Figure 3: Forward weight gradients alignment across layers of a VGG9-like CNN trained for 50 epochs on CIFAR-100. Each curve shows the cosine similarity between the instantaneous forward-weight gradient produced by DKP and DKP-PC algorithms, compared to the one computed with BP. All gradients exclude weight decay and momentum and are smoothed using an exponential moving average with a window of 100 batches. DKP (brown) displays positive but slow alignment with BP, progressively deteriorating with increasing distance from the output layer. DKP-PC (yellow) gradient is computed as sum of the gradients resulting from DKP and PC stages. It achieves consistently faster, higher, and more stable alignment across all layers compared to standard DKP. The light blue curve shows that disabling the PC forward-weight update in DKP-PC causes alignment to collapse in all layers, confirming its role in injecting alignment information into the forward weights. The blue curve, obtained by disabling the feedback-weight update in DKP-PC, demonstrates that the alignment and regularization terms introduced by the PC stage also improve the update of the feedback matrices, resulting in worse alignment when disabled.

alignment information in the forward weights through the updated neural activity. Moreover, the improvement in DKP-PC alignment also arises from the influence of the alignment and regularization terms in Eq. (64) on the feedback matrix update via Eq. (66). This yields a better-aligned update and again compensates for the distortion introduced by the Moore–Penrose pseudoinverse. This claim is empirically supported by the results for DKP-PC without DKP feedback weight update (blue curves). Consistent with expectations, alignment decreases relative to DKP-PC, exhibiting a slower and less effective value in every layer.

Through this analysis, we complement the preceding theoretical subsection by empirically demonstrating the synergy between the DKP and PC stages in DKP-PC. We show that DKP not only helps PC overcome two of its main limitations, exponential error decay and error propagation delay, but that PC, in turn, acts as a regularizer for the DKP update, improving gradient alignment with BP across all layers and yielding a more effective learning algorithm. These results also reveal a possible alternative interpretation of DKP-PC: it may be viewed both as an acceleration mechanism for training PC networks and, alternatively, as a method to enhance alignment in feedback-alignment approaches. Furthermore, DKP-PC achieves these improvements without compromising performance or the locality of computations and updates, thereby preserving the hardware-friendly properties of both PC and DKP.

### A.4.3 EMPIRICAL ANALYSIS OF THE INFERENCE PHASE

In this section, we empirically analyse the inference phase of the DKP-PC algorithm and its incremental version, denoted iDKP-PC. DKP-PC performs the update of the forward weights, given by

Eq. (15) at the end of the inference phase, after optimizing the neural activity following Eq. (14). In contrast, as introduced in Salvatori et al. (2024), iPC performs an update of the forward weights after every step of the neural activity optimization. The same concept can be applied to DKP-PC under a multiple–inference–steps regime (as opposed to the single-step regime considered in the main text of the paper). Indeed, once the forward weights are perturbed by the DKP update, they can be further optimized after each neural activity update, exactly as in iPC. The key difference is that an error signal is already available at the very beginning of the inference phase, both for the neural activity and for the forward weights.

Figure 4 compares the energy of four different four-layer MLP models trained on a Fashion-MNIST batch and averaged over 10 trials, while varying the magnitude of the neural activity learning rate. The curves depict the energy evolution during the inference phase for DKP-PC (blue), iDKP-PC (light blue), PC (brown), and iPC (yellow). Both DKP-PC and iDKP-PC start from a higher energy than PC and iPC, as mitigating the error-delay problem provides an error term at every layer, leading to larger energy values. In both cases, the incremental algorithms decrease toward a lower minimum than the non-incremental ones. This follows from the additional degree of freedom introduced by updating the forward parameters during inference. Interestingly, DKP-PC and iDKP-PC respectively converge to similar energy values than PC and iPC, suggesting that after several neural activity updates, the gradient term from PC in Eq. (62) dominates the dynamics, driving the networks toward similar low-energy regions. However, unlike standard PC, DKP-PC requires more optimization steps to reach these low energy values, likely due to both the larger error present at every layer and the presence of two distinct driving forces in the neural activity update: the standard PC gradient and the term implicitly introduced by DKP. Notably, the incremental version of DKP-PC demonstrates a convergence speed similar to that of standard iPC instead. This suggests that allowing forward parameters to update during inference compensates for the additional complexity introduced by DKP, effectively stabilizing the dynamics and enabling the network to exploit the immediate availability of error signals.

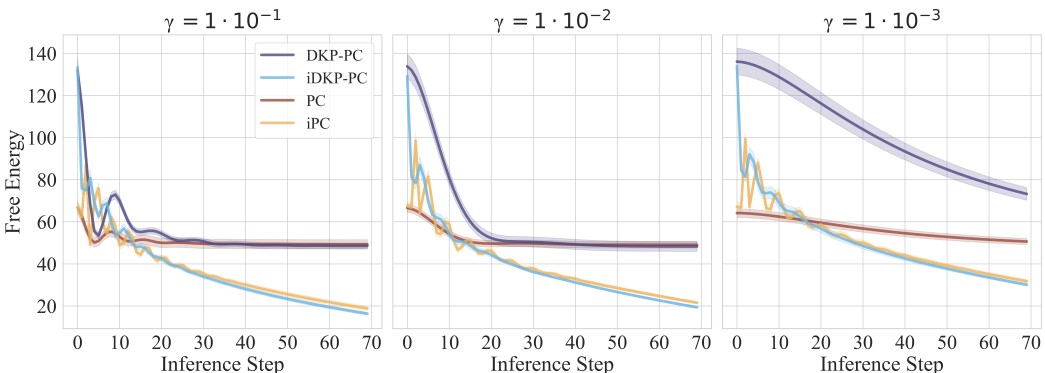

Figure 4: Energy evolution of four-layer MLP networks on a Fashion-MNIST batch for three different neural activity learning rate magnitudes. DKP-PC and its incremental variant are shown in blue and light blue, respectively, while standard PC and iPC are represented in brown and yellow. Both DKP-PC variants start from higher energy values due to the immediate error term at every layer, and converge to levels similar to those of the standard PC and iPC networks. Although DKP-PC exhibits slower convergence due to the additional terms in the neural activity dynamics, its incremental version equals the convergence speed of iPC across all evaluated learning rates, suggesting that updating forward parameters during inference effectively compensates for the additional complexity introduced by DKP.

While in principle the neural activity optimization in PC can be run until full minimization of the FE, in practice it is performed for a finite number of steps (Pinchetti et al., 2024), typically exceeding the network's depth to achieve the best results (Goemaere et al., 2025). Figure 5 presents the corresponding behaviour for DKP-PC and iDKP-PC by reporting their test accuracy as a function of the total number of inference steps, with boxplots showing the distribution across 30 trials. The evaluated networks are the same as for Figure 4. Both algorithms exhibit a positive correlation between the number of neural activity optimization steps per batch and the final test accuracy, with

higher numbers of inference steps yielding the best results in both cases. This also highlights a fundamental trade-off for DKP-PC networks, consistently with PC networks, between training time and performance. Nevertheless, as also noted in the main text, our proposed DKP-PC approach enables a better exploitation of the trade-off, as even a single inference step is sufficient to achieve results that surpass both baselines and advanced PC variants.

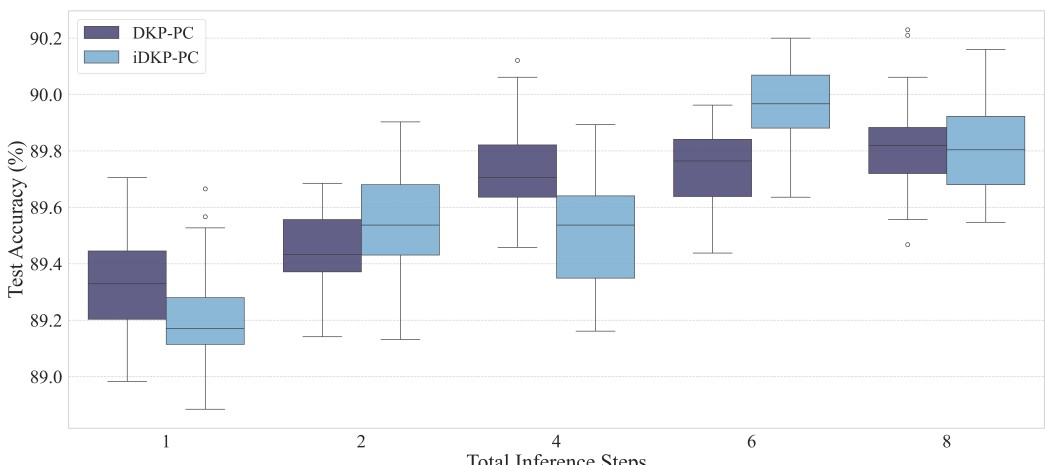

Figure 5: Test accuracy distributions over 30 trials are shown as a function of the total number of neural activity optimization steps. Blue boxplots correspond to a four-layer MLP trained on Fashion-MNIST with DKP-PC, while light blue boxplots show the same architecture trained with iDKP-PC. In line with PC theory, both methods display a positive correlation between the number of optimization steps and the final test accuracy.

## A.5 DETAILS OF CLASSIFICATION EXPERIMENTS

*Models and datasets* – The MLP models are evaluated on the MNIST and Fashion-MNIST datasets, which contain 28×28 grayscale images and comprise 60k training and 10k test samples across 10 classes. The CNN architectures are evaluated on CIFAR-10, CIFAR-100, and Tiny ImageNet. CIFAR-10 and CIFAR-100 consist of 32×32 RGB images with 50k training and 10k test samples for 10 and 100 classes, respectively. Tiny ImageNet contains 200 classes with 100k training and 10k validation images of size 64×64. The MLPs use two hidden layers with 128 units each. The VGG-like CNNs include six convolutional layers and either one or three fully-connected layers, depending on the variant. To ensure comparability with prior work and to support future benchmarking efforts, we adopt the same architectures as Pinchetti et al. (2024). The full architectural specifications are reported in Table 3. For completeness and to ensure reproducibility, Table 4 reports the optimal DKP-PC hyperparameters identified through our hyperparameter search, which were successively used to evaluate the model and obtain the results presented in Section 4.

Table 3: Architectural details. FC size refers to the number of units in the fully-connected layers after the convolutional ones (if any).

|  | MLP | VGG-7 | VGG-9 |
|---|---|---|---|
| Conv. channels | – | [128x2, 256x2, 512x2] | [128x2, 256x2, 512x2] |
| Kernel sizes | – | [3, 3, 3, 3, 3, 3] | [3, 3, 3, 3, 3, 3] |
| Strides | – | [1, 1, 1, 1, 1, 1] | [1, 1, 1, 1, 1, 1] |
| Paddings | – | [1, 1, 1, 0, 1, 0] | [1, 1, 1, 1, 1, 1] |
| Pool window | – | $2 \times 2$ | $2 \times 2$ |
| Pool stride | – | 2 | 2 |
| FC size | [128, 128, output] | [output] | [4096, 4096, output] |

Table 4: Hyperparameters employed for the DKP-PC networks in our experiments.

|  | MLP | | VGG-7 | | VGG-9 | | |
|  | MNIST | FMNIST | CIFAR-10 | CIFAR-100 | CIFAR-10 | CIFAR-100 | Tiny ImageNet |
| --- | --- | --- | --- | --- | --- | --- | --- |
| activation | gelu | gelu | gelu | tanh | leaky | leaky | gelu |
| fw-lr | 4.616e−4 | 5.254e−4 | 1.458e−4 | 2.482e−4 | 1.609e−4 | 1.602e−4 | 7.373e−5 |
| fw-decay | 3.737e−2 | 2.744e−5 | 3.626e−4 | 9.664e−2 | 5.271e−2 | 1.040e−2 | 2.893e−5 |
| fw-opt | adamw | adamw | adam | adam | adam | adam | adamw |
| i-lr | 1.068e−3 | 8.297e−1 | 5.655e−2 | 1.036e−2 | 1.113e−3 | 1.169e−2 | 3.136e−3 |
| i-mom | 0 | 0 | 0 | 0 | 0 | 0 | 0 |
| i-steps | 1 | 1 | 1 | 1 | 1 | 1 | 1 |
| fb-init | ka-unif. | ka-unif. | orthog. | ka-norm. | ka-unif. | xav-unif. | orthog. |
| fb-lr | 3.024e−5 | 4.702e−5 | 1.533e−3 | 1.333e−3 | 1.664e−3 | 9.405e−4 | 2.839e−4 |
| fb-decay | 2.446e−3 | 2.744e−5 | 5.215e−5 | 4.406e−5 | 1.099e−4 | 1.040e−2 | 4.656e−5 |
| fb-opt | adamw | nadam | adamw | adamw | adam | nadam | adam |
| fb-gamma | 0.99975 | 0.9995 | 1 | 0.99995 | 0.9999 | 0.9995 | 0.99925 |

*Training setup –* Consistently with the work of Pinchetti et al. (2024), MLPs are trained for 25 epochs and CNNs for 50 epochs, using a batch size of 128. Data augmentation is applied in the CNN experiments. For CIFAR-10 and CIFAR-100, images are randomly cropped to 32×32 pixels with 4-pixel padding during training. For Tiny ImageNet, random crops of 56×56 pixels are used during training without padding, while the test set is evaluated using centered crops of 56×56 pixels, also without padding. The forward weights' learning rate is updated using a warmup-cosine-annealing scheduler without restarts. The optimizers considered include Adam and AdamW. Feedback connections are trained with a separate optimizer than the forward weights, using an exponentially decaying learning rate updated per batch via an exponential learning rate scheduler, with the update parameter fb-gamma reported in Table 4. Different feedback initializations are explored during the hyperparameter search: Xavier-uniform/normal (Glorot & Bengio, 2010), Kaiming-uniform/normal (He et al., 2015), and orthogonal (Saxe et al., 2013). Feedback optimizers include Adam, AdamW, and Nadam (Adam et al., 2014; Dozat, 2016; Loshchilov & Hutter, 2017). All additional experimental details are available on our GitHub repository [link will be included after double-blinded revision].

## A.6 COMPUTATIONAL TRADE-OFFS

Resource consumption experiments focus on latency and floating-point operations (FLOPs), evaluated on both an MLP and a CNN. For the MLP, experiments are conducted on a network with 256 units per layer using a single sample from MNIST, whereas for the CNN, a VGG-like model with 64-channel $3 \times 3$ convolutions is evaluated on a single sample from CIFAR-10. Latency is defined as the time required for a complete parameter update, including feedforward initialization and the update of forward and feedback matrices (when applicable). Computational cost is estimated by counting FLOPs in forward and backward passes, restricted to core MAC operations, with 1 MAC = 2 FLOPs[4].

The first row of Figure 6 illustrates the differences in training time, expressed in milliseconds, across the various models. Training time includes the forward pass and a complete backward pass, encompassing both forward and feedback weight updates, if applicable. The forward pass contribution is indicated by a red dashed line. The fastest algorithm is DKP, owing to the reduced dimensionality introduced in its update in Eq. (24). BP is the second fastest algorithm, followed by DKP-PC. The latter, as well as the other local algorithms, have been evaluated in sequential mode, and therefore their parallelization potential is not considered in this section. Nevertheless, this already highlights the speed-up of DKP-PC compared to standard PC or iPC, as it consistently requires less time to fully update its parameters for both the MLP and CNN.

---

[4]Variable contributions, such as activation function FLOPs, are excluded as they depend on the specific non-linearity. Accordingly, the reported FLOPs represent a lower-bound estimate of the actual computational cost

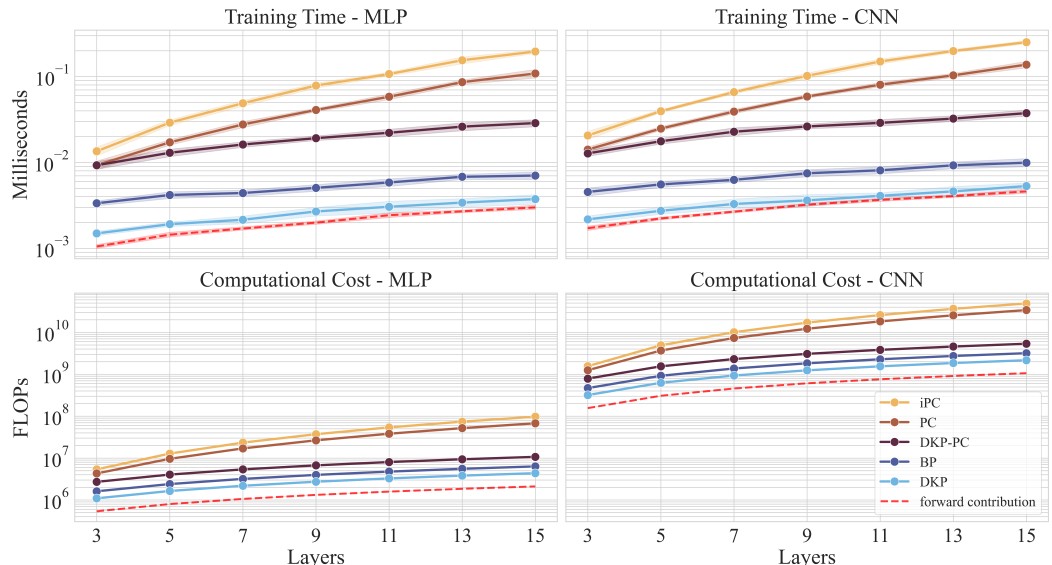

Figure 6: The first row compares the training time on a logarithmic scale, measured as the sum of the forward pass and the complete parameter updates, with the contribution of the forward pass illustrated as a red dashed line. Results are averaged over 20 samples. The second row compares the minimum FLOPs requirements, estimated from the core MAC operations. In all plots, for both PC and iPC, the number of inference-phase steps is assumed to equal the network depth. In contrast, for DKP-PC, only a single step is considered, as it has been empirically demonstrated to be enough to achieve comparable performance.

The second row of Figure 6 compares the minimum FLOPs requirements across algorithms. DKP emerges as the most efficient method in architectures where hidden layers exceed the output layer in size. In this scenario, computing the intermediate error term $\delta_\ell$ only requires multiplying the output error $\delta_L \in \mathbb{R}^{d_L}$ by the random matrix $\Psi_\ell \in \mathbb{R}^{d_\ell \times d_L}$, which entails fewer MAC operations than BP. In contrast, BP requires multiplying $\Theta_{\ell+1}^\top \in \mathbb{R}^{d_{\ell+1} \times d_{\ell+2}}$ with the higher-layer error $\delta_{\ell+2} \in \mathbb{R}^{d_{\ell+2}}$, typically with $d_L < d_\ell$ for all $\ell \in \{0, \ldots, L-1\}$. BP is the second most efficient algorithm overall, followed in order by DKP-PC, PC, and iPC. The logarithmic scale of the plot highlights the growth in computational complexity for PC and iPC as depth increases, since both the minimal number of inference-phase steps and the number of multiple matrix–vector multiplications per inference-phase step increase with depth. DKP-PC scales better than PC and iPC as it requires only one inference-phase step to match or surpass their accuracy, achieving nearly an order of magnitude fewer FLOPs, thereby underscoring its efficiency advantage.

### A.7 ANALYSIS OF PARALLEL EXECUTION

While a fully-parallel implementation of PC is theoretically possible, it has so far been practically limited by the signal error delay and exponential decay problems (Zahid et al., 2023; Pinchetti et al., 2024; Goemaere et al., 2025) detailed in Section 3.1. Here, we discuss how DKP-PC overcomes these limitations, yielding an algorithm capable of achieving lower training latency than BP.

Referring to Algorithm 1 and excluding the forward initialization, which incurs the same computational cost as BP, we first consider the *Direct Feedback Alignment update* (1). Since each forward weight update depends only on the local neural activity and the error signal propagated from the output layer via the corresponding feedback matrix, all updates can be executed in parallel, reducing the time complexity of this phase from $\mathcal{O}(L)$ to $\mathcal{O}(1)$. In the subsequent *Inference phase* (2), which is typically executed over multiple steps ($T \geq L$) in PC, we empirically demonstrate in Table 1 that a single step suffices to achieve accuracy comparable to or exceeding that of standard PC, reducing the time complexity from $\mathcal{O}(T)$ to $\mathcal{O}(1)$. Within this phase, updates of error neurons and neural activities are also parallelizable, as each relies solely on locally available information, resulting in

$\mathcal{O}(1)$ time complexity. For completeness, we highlight that synchronization is still required, as error terms must be computed before updating neural activities. Successively, the *Learning phase* (3) and *Direct Kolen-Pollack update* (4) are executed sequentially, though they are independent and can be performed simultaneously. While these phases respectively have to iterate over all $L - 1$ forward and backward parameter matrices, their computations are entirely local and hence can be executed in parallel across layers, reducing the time complexity from $\mathcal{O}(L)$ to $\mathcal{O}(1)$.

In summary, DKP-PC consists of four phases, each theoretically fully parallelizable with time complexity $\mathcal{O}(1)$. Each phase blocks the next, except for the final two, which may run concurrently. Consequently, DKP-PC's time complexity does not grow with network depth, unlike BP, whose time complexity scales linearly as $\mathcal{O}(L)$. For sufficiently deep networks, we claim that the overall training time of DKP-PC will be significantly lower than that of BP. In practice, however, this advantage is challenged by existing hardware, which is heavily optimized for BP, and by the synchronization overhead inherent in software-based parallelization. These limitations could nevertheless be overcome by custom hardware accelerators designed to fully exploit DKP-PC's parallelizability.

### A.8 Large Language Models usage disclosure

Large Language Models (LLMs) were employed only for language polishing, such as grammar and phrasing refinement. They were not used for content generation, results analysis, or methodological development.

