# OpenReview forum: "Accelerated Predictive Coding Networks via Direct Kolen–Pollack Feedback Alignment"
_ICLR.cc/2026/Conference — Submitted to ICLR 2026_

### Official Review · Reviewer_9Pmy · 2025-10-23

**Soundness:** 3
**Presentation:** 2
**Contribution:** 2
**Rating:** 4
**Confidence:** 5

**Summary:**

This work merges successfully two important learning algorithms for neural networks, that are predictive coding and direct (Kolen–Pollack) feedback alignment. The resulting algorithm is faster and better than standard PC, but a little more memory heavy, due to the feedback tensors. However, this is marginal compared to the improved performance in terms of test accuracy and running time.

**Strengths:**

It tackles an important problem in the field of "alternative training algorithms", that is the efficiency of the simulations. The results are quite nice. For me, the most interesting 'reading' of this work, is that it proposes a much faster PC model that is able to reach performance that are as good as the original algorithm.

**Weaknesses:**

Despite of the fact that the results are nice, I have some concerns on how the evaluation was performed, in the scale, and on the fact that a larger study could be implemented. I believe that such concern do not allow the work to be "complete" yet, and it would be nice to implement them. The concerns are:

1) You compare the results of your method against the numbers presented in Pinchetti et al. 2024. While that one is a good reference, you have not reported and compared against the test accuracies computed using centered or negative nudging, that are much better.

1b) It would be nice to have a table reporting the performance of DKP-PC with centered and negative nudging. This is the method that in Pinchetti et al. has achieved the best results, why have you decided not to include it?

2) Similarly, why don't you implement an 'iPC' version of DKP? It should be straightforward to do so, and it also has the potential of working better.

3) How many iterations of the PC algorithm do you run? I would be interested in looking at plots that show the energy decreasing from t=1 (so, after the backward maps have updated the errors) to t=T, where T is a large, fixed number of iterations. I would be curious in confronting that against the curves we have for PC and iPC. I would also be interested in looking at the changes in test accuracy over T, which you do not report.

I see that you have figure 2, but it still leaves multiple questions open: why is the error so large? How nicely does it decrease over time? Does it force you to use very small learning rates?

4) Despite the nice efficiency results, the test accuracies are not as good as the ones obtained via centered nudging, or more recent implementations of PC that make it work on larger and deeper models (https://arxiv.org/abs/2506.23800). We could argue whether or not this is acceptable, given the advantage in terms of running time, and I believe it is: having faster models is a good contribution. However, I feel that the authors have not placed the right effort into pushing the limits of the model they are proposing.

**Questions:**

I would be curious in looking at the hyperparameters you have used to achieve the results (Such as learning rates for the x's, and the number of iterations T). However, in the supplementary material such details are omitted, and referred to the github page (which is not provided). I'd like to see a section explaining what worked and what didn't, which combinations you have tried -- so, the search space --, and the best performing ones.

More importantly: I believe the paper to be based on a nice idea, that seems to work fine. However, there are many more experiments that could be performed that would dramatically improve the quality of the paper, as well as the results (See the 4 points I made above for ideas). I'd like to see them implemented before proposing acceptance.

---

> ### Author Response · Authors · 2025-11-24
>
> We thank the reviewer for the valuable points raised. We first address the weaknesses below, in the order in which they were raised:
>
> - **Nudging**: We thank the reviewer for this comment. Following a coarse-grained hyperparameter search, we did not obtain any noticeable and consistent improvement when combining DKP-PC with nudging. Due to computational and time capacity constraints, expanding the HP search and exploring this aspect further would come at the expense of the other suggestions that more directly align with strengthening our central claims along the efficiency axis. To address this comment, we will thus (i) include a dedicated appendix section with a comparison between nudging PC and nudging DKP-PC, for the sake of completeness, and (ii) we have also mentioned this promising direction in the future work section in lines 466-470, as follows: ‘Future work could also focus on a tailored integration of DKP with advanced PC variants, such as nudging PC based on equilibrium propagation \citep{scellier2017equilibrium, scellier2023energy, pinchetti2024benchmarking}, combining their dynamics with the DKP learning rules for both forward and feedback weights. This integration could allow the different formulations to complement each other and further reduce the performance gap with BP.’
>
> ---
>
> - **IDKPPC**: We thank the reviewer for raising this point. DKP-PC can indeed be adapted into an incremental variant. Initially, since our experiments show that a single neural activity optimization step already yields strong performance, we did not pursue this direction, which would additionally diminish the computational advantage of standard DKP-PC. We nevertheless agree this was missing in the manuscript, we will thus add a dedicated appendix section (see also next point).
>
> ---
>
> - **Analysis**: We agree with the reviewer that the proposed analyses would provide valuable additional insights. We will include in the appendix a section examining both the energy and the accuracy trends beyond a single neural activity optimization step, comparing DKP-PC with PC and iPC, as well as with iDKP-PC, thereby also addressing the reviewer’s previous point. We will follow up further on these experiments by the rebuttal deadline.
>
> ---
>
> - **Comment**: We sincerely thank the reviewer for understanding and acknowledging the direction taken in our work and for appreciating its contribution to the PC field. Indeed, we focused on a different performance axis compared to the usual effort in the PC literature on improving scalability. While our approach does not scale as well as the latest advancements in PC, we achieve state-of-the-art performance in terms of training speed, and we also unlock PC's potential emerging from the parallelizability of its local update rules, an appealing property for hardware accelerators (as also mentioned in the suggested reference). We hope that the additional experiments linked to the suggestions above, beyond strengthening the manuscript, will address the reviewer’s questions. We remain available for further discussion.
>
> ---
>
> Now, regarding the question raised by the reviewer:
>
> - **On hyperparameter transparency**: We agree with the reviewer that these details are crucial, both for understanding the approach and ensuring reproducibility. While we did not include the GitHub link to avoid invalidating the double-blind review process, it will be publicly released once permitted. Nevertheless, to address this point, we will extend the "Details of Classification Experiments" section of our paper to include the best hyperparameters for each model, along with full architectural details and the ranges of hyperparameters explored.
>
> ---
>
> We hope that our responses have addressed the reviewer’s questions and that the additional experiments we plan to include before the rebuttal deadline will address any remaining concerns. We sincerely thank the reviewer for their constructive feedback and remain available for further discussion, if needed.

---

> > ### Comment · Reviewer_9Pmy · 2025-11-27
> >
> > Thank you for your answer. I will then comment the updated manuscript.

---

> ### Author Response · Authors · 2025-12-03
> **Follow-up**
>
> We would like to follow up on the rebuttal above, as we have now addressed all the remaining points of the feedback provided, for which we are grateful.
> Specifically, we have included centered nudging (CN)-PC results in our benchmarking table and evaluated our DKP-PC models against them. This includes experiments on the Tiny ImageNet dataset, where DKP-PC outperforms CN-PC. We included CN-PC as it can be considered a generalization of positive and negative nudging. This not only aligns with the reviewer’s fourth point, but now also highlights the scalability of DKP-PC from an accuracy point of view: “Despite the nice efficiency results, the test accuracies are not as good as the ones obtained via centered nudging [...]. We could argue whether or not this is acceptable, given the advantage in terms of running time, and I believe it is.”
>
>
> Regarding the incremental version of DKP-PC, we have added a new section in Appendix A.4 focusing on the analysis of the inference phase. Here, we show that the energies of both DKP-PC and iDKP-PC start from higher values (as expected, given the mitigation of the error delay typical of (i)PC networks) and then converge to levels similar to those of (i)PC networks. We also illustrate how network accuracy increases as a function of inference steps, highlighting a trade-off between performance and training speed. Unlike standard (i)PC networks, DKP-PC can initiate this trade-off with just a single inference step.
>
>
> Finally, we have added detailed information on the network architectures used in the evaluation, along with an extensive table containing all hyperparameter details (pending the public release of our GitHub repository, which we withheld to maintain anonymity during the review process).
>
> We hope that these additions fully address all questions and concerns raised by the reviewer, and we thank the reviewer again for the insightful feedback.

---

### Official Review · Reviewer_V7AK · 2025-10-31

**Soundness:** 3
**Presentation:** 3
**Contribution:** 2
**Rating:** 2
**Confidence:** 4

**Summary:**

The paper proposes DKP-PC, a predictive-coding variant that injects learnable direct feedback connections from the output layer to every hidden layer, ensuring that error signals are present everywhere at the first inference step, thereby addressing the typical PC issues of error delay and exponential decay. This makes PC effectively depth-independent in time, and on CIFAR/VGG benchmarks, DKP-PC consistently outperforms standard PC/iPC while staying below full backpropagation.

**Strengths:**

- The manuscript is clearly structured and accessible: it introduces the predictive-coding bottleneck, formulates the DKP-style remedy, and presents empirical results.
- Moreover, to the best of my knowledge, it constitutes the first explicit attempt to integrate feedback-alignment (DKP) mechanisms with predictive coding for the specific purpose of eliminating PC’s time–depth coupling.

**Weaknesses:**

1. **Limited novelty**
The core contribution can be viewed as a relatively simple composition of two well-established ideas—predictive coding for local inference and DKP/feedback alignment for direct error delivery—without introducing a clearly new theoretical principle or a substantially different training regime. As such, part of the contribution lies more in integration than in genuine algorithmic innovation.

2. **Performance limitations**
When contrasted with recent PC-acceleration efforts (e.g., https://openreview.net/pdf?id=s3E08R4AMK), the reported results do not consistently outperform the best published performance on comparable settings, which weakens the empirical claim that the proposed mechanism is the most effective way to mitigate error delay/decay. A short, head-to-head comparison on the same architecture/dataset/protocol would be needed to substantiate the performance advantage.

3. **Insufficient experimental diversity**
The evaluation is focused on a narrow set of convolutional backbones and CIFAR-scale datasets, making it challenging to assess whether the method scales to architectures with different connectivity patterns (e.g., ResNet) or to higher-resolution, more challenging datasets, such as Tiny-ImageNet. Broader model and dataset coverage would make the generality of the proposed approach more convincing.

**Questions:**

1. Could you clarify why DKP was chosen over more recent state-of-the-art feedback-alignment methods, and what concrete advantages it provides in the predictive-coding setting?

2. Is there a possibility for designing a novel feedback-alignment mechanism specifically tailored to predictive coding, so that it better addresses error delay and depth-dependent attenuation and potentially closes more of the gap to backpropagation?

3.  A clearer discussion of whether the joint use of PC and DFA/DKP remains biologically plausible would be helpful.

---

> ### Author Response · Authors · 2025-11-24
> **Reply to the weaknesses mentioned by the reviewer**
>
> We thank the reviewer for the valuable points raised. We first address the weaknesses below, in the order in which they were raised:
>
> - **On novelty**: We would first like to emphasize that many works have their core novelty in addressing open challenges of existing algorithms, rather than in introducing fundamentally new theoretical principles. A relevant example is the one of incremental Predictive Coding (iPC) [ICLR 2024, https://arxiv.org/pdf/2212.00720], which introduced a simple yet effective idea to improve performance and address the error delay problem of PC: changing the temporal scheduling of the update rule. Another example lies in Direct Kollen-Pollack (DKP), which could also be seen as just a combination of existing KP and direct feedback alignment (DFA), yet is a fundamentally valuable contribution. Similarly, while our work relies on two well-established algorithms, its novelty does not reside exclusively in their combination. Indeed, we introduce DKP as a way to address one of the main limitations of PC networks, namely the need to perform a slow and expensive nested optimization on the neural activity. We also analyze how, unlike iPC, this formulation allows for fully parallel training regardless of the batch size, thereby overcoming for the first time the lack of parallelizability for mini-batch training of both PC and iPC, which was a key barrier to the adoption of PC networks. Importantly, to provide a strong theoretical grounding for the usage of DKP, we also extend the empirical analysis in the original study, showing why DKP has a better alignment with backpropagation than when using fixed random connections (as with the more common DFA algorithm). We believe the relevance of addressing this challenge, as well as the supporting empirical and theoretical developments, makes our contribution novel and valuable.
>
> ---
>
> - **On performance and actual claims**:  The reviewer writes: ‘When contrasted with recent PC-acceleration efforts [...], the reported results do not consistently outperform the best published performance on comparable settings, which weakens the empirical claim that the proposed mechanism is the most effective way to mitigate error delay/decay.’ The work referred to by the reviewer links to a study on PC that focuses on a more advanced error weighting strategy, in order to mitigate the imbalanced energy distribution in PC. It does not address training acceleration nor the unlocking of the theoretical parallelizability of PC networks, and is thus not related to the claims and contributions of our work. Importantly, the existence of other works that focus on improving different aspects of PC, such as classification accuracy, does not invalidate our claims that the proposed DKP-PC algorithm is the most effective PC variant for addressing error delay, thereby unlocking the acceleration of PC networks via parallelization. To the best of our knowledge, the only variant that targets the same objective is iPC (though only for full-batch training), which we explicitly discuss and benchmark in the paper. In case the reviewer is aware of any other study that also targets PC acceleration and yet achieves better performance, we would like to kindly ask the reviewer to provide us with the reference so that we can include it in our manuscript and discuss it.
>
> ---
>
> - **On experimental diversity**: We agree with the reviewer that incorporating experiments with larger models and more complex datasets would add value to the paper and allow for a more thorough analysis of the proposed algorithm's limitations in terms of classification accuracy. Since ResNet architectures are known to be unstable and harder to train in the context of PC (see the caption of Table 1 in  Section 5 of https://arxiv.org/pdf/2505.20137), we will prioritize adding tinyImageNet experiments within the rebuttal time frame, in order to address the weakness identified by the reviewer. We will follow up on this by the rebuttal deadline.

---

> ### Author Response · Authors · 2025-11-24
> **Reply to the questions raised by the reviewer**
>
> Due to the character limit, we have to split the discussion into two comments. We now address the questions raised by the reviewer:
>
> 1. **On the choice of DKP over more recent state-of-the-art feedback-alignment methods**: To the best of our knowledge, DKP represents one of the most recent advancements in feedback alignment methods. Its selection, as stated in the paper, is motivated by its ability to (i) enable parallel error generation at every layer, (ii) align with BP thanks to the update of the backward weights, and (iii) keep a fully local update rule. If the reviewer is aware of any additional feedback alignment works that meet these three criteria, we would like to kindly ask for the references in order to include and evaluate them.
>
>
> ---
>
> 2. **On the possibility of designing a novel feedback-alignment mechanism specifically tailored to predictive coding**: This is a very insightful question, and we welcome the opportunity to use it to underscore the broader importance of our current study. Our work not only presents a novel approach to address PC's error delay problem, but indeed also paves the way for a novel area of research within the PC field: the principled combination of PC with feedback-alignment mechanisms to enhance both efficiency and performance. Regarding the specific point raised, we do think that it should be possible to design a DFA-like method tailored specifically for PC. While our current approach updates the forward weights, a similar effect could be achieved by directly updating the neural activity during the inference phase, adjusting each layer’s activity to minimize a joint objective involving both the Free Energy and the global output error. This strategy could reduce computational cost, since neural activities have lower dimensionality than weight matrices, and may lead to more informative updates of both neural activities and forward weights. As our current contribution provides the necessary preliminary evidence and conceptual foundation, we have emphasized this research direction in the future work section of the manuscript, at lines 454-461, as follows: 'Furthermore, as feedback matrices introduce memory overhead, sparsity and quantization of feedback weights should be explored [...]. Lastly, this novel combination of feedback-alignment methods and PC might pave the way for a new class of algorithms focused on exploiting the synergy between the two frameworks and leveraging their specific dynamics. An interesting research direction is to directly use the feedback information to perturb the neural activity dynamics, without relying on a preliminary weight update step. This approach could lead to new and more efficient local update rules for the neural activity dynamics, resulting in faster and more effective algorithms.'
>
> ---
>
> 3. **On the biological plausibility of the joint use of PC and DFA/DKP**: Interestingly, DKP-PC does align with biologically plausible mechanisms. Indeed, beyond the well-known alignment of PC with several brain theories (https://arxiv.org/pdf/2112.10048, https://pubmed.ncbi.nlm.nih.gov/15937014/, https://www.sciencedirect.com/science/article/pii/S092842570600060X), there is neuroscientific evidence for projections from V1 to different higher areas of the visual cortex, conceptually aligning with the DKP framework (see page 31, https://www.cns.nyu.edu/~tony/vns/readings/felleman-vanessen-1991.pdf). Moreover, although the sequential nature of the algorithm may appear to challenge strict biological plausibility, each stage relies exclusively on locally available variables, preserving the locality of computation throughout the entire learning process, which is a hallmark of bio-plausible learning. We thank the reviewer for raising this point, and we refer to lines 355-360 of the manuscript, where we have included the previous statement as follows:  'With DKP-PC, we introduce the first PC variant that is fully parallelizable. Although its sequential formulation between stages may appear to challenge parallelizability and strict biological plausibility, each stage relies exclusively on locally available variables. This enables full parallelization across layers and preserves the locality of computation throughout the entire learning process, which is a hallmark of biologically plausible learning. As a result, the backward time complexity of the network is reduced from [...].'
>
> ---
>
> We hope the reviewer’s questions have been answered and will follow up regarding tinyImageNet experiments. We are hoping to receive pointers to the missing references that the reviewer mentioned, and we remain available to engage in further discussion.

---

> ### Author Response · Authors · 2025-12-03
> **Follow-up**
>
> We would like to follow up on the rebuttal above, as we now have included experiments on Tiny ImageNet using our most complex evaluated network (VGG9), demonstrating that our approach outperforms all other evaluated local algorithms in this new scenario.
> Additionally, as the reviewer pointed out a need for additional theoretical analyses and lack of novelty, we would like to bring to the reviewer’s attention Appendix A.4, where we provide both empirical and theoretical analyses of how DKP-PC neural activity dynamics differ from those of standard PC networks, and how these differences impact gradient alignment and test accuracy.
>
> We thank the reviewer for these suggestions.

---

### Official Review · Reviewer_mjHt · 2025-10-31

**Soundness:** 2
**Presentation:** 4
**Contribution:** 3
**Rating:** 6
**Confidence:** 4

**Summary:**

The authors propose combining ideas from feedback alignment in order to overcome known issues with Predictive Coding.  They demonstrate empirically their proposed algorithm performs better than both PC and DKP in terms of accuracy over a range of datasets and models.

**Strengths:**

Using ideas from feedback alignment to solve the signal propagation issues with PC is a novel idea and addresses important problems for potential hardware implementations.
The authors report significant peformance benefits over traditional predictive coding and KP-alignment.
The clarity of the paper is very good.

**Weaknesses:**

The authors give valid justifications for introducing feedback connections to improve problems with PC. However, the authors do not justify (other than benchmark) why the single inference step would be improvement over pure feedback alignment.

This is hard to reason about due to the (prima facie strange) choice of updating the weights twice per forward pass.  Indeed it is not clear what the inference step is minimising - at worst it may just act as a damping factor on the DKP update (since the layer wise errors are zero exactly when they reflect the new weights).

Some argument as to why PC inference steps improve DKP create a much stronger paper, talking directly to both communities.

Minimally, more space in the paper should be dedicated to justifying the use of inference steps. It would be great to see some empirical analysis to this effect such as:

Checking the alignment of the weight updates with the gradient update, with and without the inference step.
Ablations showing the effect of multiple inference steps and/or the learning rate used in inference.
Or a theoretical justification of why inference should help.  Without any of this we are  left to trust the single table of numbers - in which case a more detailed description of all hyperparameter and training details should be given.

Secondly, if the authors truly believe their formal arguments extend Webster’s results is a contribution it should be included in the main paper rather than the appendix. Perhaps, there is the potential to explicitly use this analysis to demonstrate the weakness of pure DKP and justify the inference step as above?

**Questions:**

Mainly Question as above:

What do the authors believe the inference step is doing?
Specific Questions:

Did the authors consider defining extra error terms in the Predictive Coding energy of the form
ϵ' _l =(ϕ_l−Ψ_l  y)^2
  and just doing typical PC updates on this (instead of the double weight update algorithm proposed)? If so what was the conclusion? — it seems when traditional errors are small this term would dominate providing early signal, which could then be corrected by PC dynamics.

---

> ### Author Response · Authors · 2025-11-27
> **Addressing the weaknesses and questions raised by the reviewer**
>
> We thank the reviewer for the constructive observations. We address each of the listed weaknesses below, following the sequence in which they were introduced:
>
> - **On the PC-DKP combination:** We thank the reviewer for raising this point. To address this weakness, we have expanded the appendix with additional experiments on the impact of the inference step, giving more insights into how it affects the network’s alignment, both theoretically and empirically. As requested, a more detailed description of all hyperparameters and training details has also been added to the relevant section. Furthermore, a section analyzing the behavior of the energy evolution and test accuracy as a function of the number of inference steps will be included. We will follow up further on the latter by the rebuttal deadline.
>
> ---
>
> - **On the DKP contribution:** We thank the reviewer and agree with this point. Therefore, we have incorporated the final approximate convergence formulation of DKP into the main paper as follows:  '[...] by mathematically showing that, under linear assumptions, the feedback matrices converge to values incorporating the recursive chain in (4), despite the dimensionality discrepancy between $\Psi_\ell$ and $\Theta_\ell$. We provide a new theoretical perspective suggesting why DKP aligns more closely with BP than DFA, demonstrating that it converges to a recursive Moore–Penrose pseudoinverse chain of the forward weights:
> \begin{equation}
> \Psi_{L-\ell} =
> \begin{cases}
>     \Theta_{L-\ell}^\top & \text{if } \ell = 1,\\
>     \Theta_{L-\ell}^\top \left( \Psi_{L-\ell+1}^\top \right)^{+} & \text{if } 1 < \ell < L.
> \end{cases}
> \end{equation}
>
> ---
>
> Now, regarding the questions raised by the reviewer:
>
> - **On the role of the inference step:** We thank the reviewer for raising this question. While the PC perspective on DKP-PC, on which the paper is focusing, is that DKP can be seen as  a way to inject information about the output loss directly into the neural activities updates, another perspective from the DKP point of view is that PC may indeed act as a regularizer on the weight update. As mentioned in the related point above, we have included a new appendix (Appendix 4) for expanding the discussion on this matter from both the theoretical and empirical points of view. We refer the reader to into the main text in lines 354-358 as follows: ‘ In contrast to standard PC, the neural activity now incorporates the information injected into the forward weights by the preliminary DKP update. In Appendix 4.1, we show theoretically that, under linear assumptions, this corresponds to enforcing alignment and regularization through the single-step neural activity update, which in turn improves the alignment of the forward and feedback weights, as further supported by empirical evidence in Appendix 4.2.’
>
> ---
>
> - **On the suggested definition and interpretation of extra error terms in the Predictive Coding energy :** We thank the reviewer for this insightful point. We had not yet explored the mentioned approach, nor more specifically the proposed formulation, but we agree that incorporating the feedback signal directly into the network’s neural activities rather than into the weights is certainly a promising direction for future work. Furthermore, this could lead to a more efficient version of DKP-PC, as the dimensionality of the neural activities is always lower than that of the weights, resulting in fewer FLOPs per overall update. We are for now prioritizing other experiments that were suggested given the tight rebuttal deadline, and meanwhile have already included this idea to the future work section as follows: 'Furthermore, as feedback matrices introduce memory overhead, sparsity and quantization of feedback weights should be explored [...]. Lastly, this novel combination of feedback-alignment methods and PC might pave the way for a new class of algorithms focused on exploiting the synergy between the two frameworks and leveraging their specific dynamics. An interesting research direction is to directly use the feedback information to perturb the neural activity dynamics, without relying on a preliminary weight update step, thereby outlining faster and more efficient local update rules for the neural activity dynamics.'
>
> We hope the reviewer’s questions have been answered and will follow up further regarding pending experiments. We remain available to engage in further discussion.

---

> ### Author Response · Authors · 2025-12-03
> **Follow-up**
>
> We would like to follow up on the rebuttal above, as we have now added an additional subsection to Appendix A.4, titled “Empirical Analysis of the Inference Phase”, where we also examine the energy behavior of DKP-PC networks. In this new subsection (Appendix A.4.3), we show that while the energy of a DKP-PC network starts at higher values (as expected due to the mitigation of exponential error decay and delay), it converges to the same levels as networks trained with standard PC. This observation aligns with our theoretical derivation, which demonstrates that the new neural activity dynamics are guided by the standard PC gradient, supplemented with additional alignment terms that drive learning in the single-step regime. In the multiple-step regime, the standard PC gradient terms progressively dominate as the inference phase continues, allowing the networks to reach an energy level comparable to that of standard PC networks.
>
> Furthermore, as requested, Appendix A.4.3 includes results for an MLP network, showing classification accuracy as a function of the number of inference steps. These results illustrate how network performance improves with additional neural activity optimization steps, consistent with the behavior observed in standard PC networks.
>
> We hope that with this additional analysis and all previously mentioned points, we have comprehensively addressed all questions and concerns raised by the reviewer, and we thank the reviewer again for the insightful feedback.

---

### Official Review · Reviewer_tRQ7 · 2025-10-31

**Soundness:** 3
**Presentation:** 4
**Contribution:** 3
**Rating:** 6
**Confidence:** 4

**Summary:**

This paper proposes a modification to predictive coding (PC) based on direct Kolen-Pollack (DKP) learning rules. The new rule, called DKP-PC, allows for fully parallelizable PC updates. Typically, PC suffers from slow propagation of error and requires many training steps to converge stably. DKP-PC works by using learned feedback weights directly from the output to each layer. After the forward pass, the weights are updated using the feedback pathway, then standard PC takes over. In practice, the feedback weights allow error to propagate immediately to all layers and reduce the number of PC convergence steps to O(1) (instead of O(L)).

**Strengths:**

1. The idea of bridging conventional feedback alignment rules and PC is novel and interesting.
2. The benefits, both theoretically and practically, are immense for enabling PC beyond niche cases.
3. Theory is correct and rigorous for the most part.

**Weaknesses:**

Limited discussion of the role feedback alignment in convergence. See Questions section for more details.

**Questions:**

Q1: It seems that feedback alignment allows for quick convergence and error correction, but PC corrects approximation errors that FA schemes cannot. Is this correct? If so, would you expand on why in the discussion?

Q2: In Eq. 37, I cannot follow the final step of the derivation. It is likely I am just missing which previous result to substitute in. Can you help clarify?

Q3: The assumptions in Appendix A.1 are too much to support the claim in Lines 170-172. I would soften this language.

Q4: Do the learning rates have to remain the same between the PC and DKP portions of the algorithm? I assume they don't, but some empirical work showing their effect on convergence in the context of Q1 would be interesting.

---

> ### Author Response · Authors · 2025-11-27
> **Reply to the questions raised by the reviewer**
>
> We thank the reviewer for the valuable questions. We address them in the order in which they were raised, and in the first response we also address the highlighted weakness:
>
> - **On the role of feedback alignment and PC in convergence and error approximation:** We thank the reviewer for this insightful question. While the interpretation of DKP-PC from a PC perspective is that the DKP-updated forward weights inject information about the output loss directly into the neural activity updates, we agree with the reviewer that PC could also be seen as a regularizer to DKP updates, affecting its alignment.  We have added Appendix 4 in the revised manuscript, which further supports this with both theoretical and empirical analyses. In the main text, we refer the reader to this appendix in lines 354-358 as follows: ‘In contrast to standard PC, the neural activity now incorporates the information injected into the forward weights by the preliminary DKP update. In Appendix 4.1, we show theoretically that, under linear assumptions, this corresponds to enforcing alignment and regularization through the single-step neural activity update, which in turn improves the alignment of the forward and feedback weights, as further supported by empirical evidence in Appendix 4.2.’
>
> ---
>
> - **On the final steps of Eq. 37:** We thank the reviewer for raising this point. In (37), we have substituted $\Psi_{L-2}$ with the terms defined in its update in (35). Under the assumption of a sufficient number of training steps, the initial values decay exponentially due to the weight decay, and the feedback matrix approaches the value given by the update in the previous step. This substitution is approximate, since an exact substitution would require infinite training time and fully converged forward weights. Nonetheless, because of the exponential decay of the initial condition and the convergence of the forward weights, this approximation remains valid under linear assumptions. Following the reviewer’s suggestion, we made this explanation more explicit in the revised manuscript, as follows: 'On the other hand, under the assumption of a sufficient number of training steps, the initial values decay exponentially due to the weight decay, and the feedback matrix approaches the value given by its update in (35), allowing us to write: [...]'
>
> ---
>
> - **On formulating the claims at lines 170-172 with regard to the assumptions in Appendix A.1:** We thank the reviewer for pointing this out, and we acknowledge that the language in the manuscript needed to be softened to reflect the assumptions in Appendix A.1. Therefore, lines 170-172 have been rewritten as follows: '[...] by mathematically showing that, under linear assumptions, the feedback matrices converge to values incorporating the recursive chain in (4), despite the dimensionality discrepancy between $\Psi_\ell$ and $\Theta_\ell$. We provide a new theoretical perspective suggesting why DKP aligns more closely with BP than DFA, demonstrating that it converges to a recursive Moore–Penrose pseudoinverse chain of the forward weights:
> \begin{equation}
> \Psi_{L-\ell} =
> \begin{cases}
>     \Theta_{L-\ell}^\top & \text{if } \ell = 1,\\
>     \Theta_{L-\ell}^\top \left( \Psi_{L-\ell+1}^\top \right)^{+} & \text{if } 1 < \ell < L.
> \end{cases}
> \end{equation}
>
> ---
>
> - **On the learning rates of DKP-PC's steps:** We thank the reviewer for this insightful question. In preliminary experiments, we found that using the same learning rate and the same optimizer for both stages yields the best performance. Moreover, decoupling the learning rates would further increase an already substantial hyperparameter space, given the feedback, forward, and neural parameters of DKP-PC. Indeed, the feedback weights are trained with a different optimizer and different hyperparameters than the forward weights, providing the network with additional degrees of freedom. If the reviewer is interested in further details on the hyperparameter, we have included the complete list in the appendix section “Detail of Classification Experiments.”
>
>
> We hope the reviewer’s questions have been answered. We remain available to engage in further discussion.

---

### Meta-Review · Area_Chair_hMQX · 2026-01-08

**Summary:**

The paper proposes DKP-PC, a predictive-coding variant that injects learnable direct feedback connections from the output layer to every hidden layer, ensuring that error signals are present everywhere at the first inference step, thereby addressing the typical PC issues of error delay and exponential decay. This makes PC effectively depth-independent in time, and on CIFAR/VGG benchmarks.

While the proposed DKP-PC provides an significant acceleration on PC learning, with a successful merging of predictive coding and feedback alignment, the main concerns leave on 1) the evaluation on different setting and comparison to different prior methods are not sufficient, 2) the performance compare to BP is still far behind, as well as the the speed, 3) the proposed DKP-PC, though effective, does not solve the key requirement - providing an alternative training algorithm to BP. This requirement, demands rather a better design for the learning mechanism to get better performance, rather than accelerating an existing method with poor performance.

Overall, although as several reviewers noted, the paper proposed a novel idea to optimize PC and show some nice results, the completeness of its empirical study is not enough. I would further argue the point 3 above to encourage the authors to explore more on the performance side.

**Reviewer Concerns:**

The novelty concerns, and some further comparisons proposed by the reviewers were addressed by the rebuttal, while the above point 2 and point 3 are still outstanding.

**Reviewer Scores:**

The original scores are 6(tRQ7), 2(V7AK), 6(mjHt), 4(9Pmy). I would expect a change to 6,4,6,4 or 6,2,6,4 if there is a full discussion since I didn't see the rebuttals fully address all the concerns.

---

### Decision · Program_Chairs · 2026-01-26

Reject